# Coupling the Community Land Model version 5.0 to the parallel data assimilation framework PDAF: Description and applications

Lukas Strebel[1,2], Heye R. Bogena[1,2], Harry Vereecken[1,2], Harrie-Jan Hendricks Franssen[1,2]

[1] Agrosphere Institute, IBG-3, Forschungszentrum Jülich GmbH, Germany

[2] Centre for High-Performance Scientific Computing in Terrestrial Systems: HPSC TerrSys, Geoverbund ABC/J, Leo-Brandt-Strasse, 52425 Jülich,

*Correspondence to*: Lukas Strebel (l.strebel@fz-juelich.de)

**Abstract.** Land surface models are important for improving our understanding of the Earth system. They are continuously
improving and becoming better in representing the different land surface processes, e.g. the Community Land Model version 5 (CLM5). Similarly, observational networks and remote sensing operations are increasingly providing more data, e.g. from new satellite products and new in-situ measurement sites, with increasingly higher quality for a range of important variables of the Earth system. For the optimal combination of land surface models and observation data, data assimilation techniques have been developed in the past decades that incorporate observations to update modelled states and parameters. The Parallel
Data Assimilation Framework (PDAF) is a software environment that enables ensemble data assimilation and simplifies the implementation of data assimilation systems in numerical models. In this study, we present the development of the new interface between PDAF and CLM5. This newly implemented coupling integrates the PDAF functionality into CLM5 by modifying the CLM5 ensemble mode to keep changes to the pre-existing parallel communication infrastructure to a minimum. Soil water content observations from an extensive in-situ measurement network in the Wüstebach catchment in Germany are
used to illustrate the application of the coupled CLM5-PDAF system. The results show overall reductions in root mean square error of soil water content from 7% up to 35% compared to simulations without data assimilation. We expect the coupled CLM5-PDAF system to provide a basis for improved regional to global land surface modelling by enabling the assimilation of globally available observational data.

## 1 Introduction

The land surface forms the interface between the atmosphere and the lithosphere and plays a crucial role in the global climate system. Therefore, land surface models (LSMs) are an important tool to progress our understanding of the Earth system. LSMs represent a wide variety of processes from energy partitioning and mass exchanges to hydrological and ecological processes. The research community has developed sophisticated parameterizations and combined them into increasingly complex and
accurate LSMs. For example, see Arora et al. (2020) for a comparison of coupled atmosphere-land surface models in terms of projected carbon concentrations and carbon feedback as part of the Coupled Model Intercomparison Project (CMIP). However,

predictions with LSMs are still affected by various important sources of uncertainty, including initial conditions, parameters, parameterization (e.g. surface and subsurface water flow), and effects of the commonly used coarse resolution of LSMs (Wood et al., 2011). Therefore, observational data are often used to improve model predictions. Here we focus on soil water content

(SWC) as it is a key variable that strongly influences the partitioning of latent and sensible heat flux as well as the partitioning of precipitation into surface runoff and infiltration (e.g. Vereecken et al. 2008). Furthermore, SWC has a strong influence on vegetation growth and modulates fire risks (e.g. Buotte et al. 2019). Humphrey et al. (2021) shows that the inter-annual variability in land carbon uptake simulated by Earth system models is driven by anomalies in temperature and vapor pressure deficit, which are controlled by soil moisture variability. However, they conclude that the partitioning between direct and

indirect soil moisture effects is more dependent on modelling approaches and that more physical and holistic modelling of the vegetation response to drought could reduce uncertainties in climate projections. A commonly used LSM is the Community Land Model (CLM) (Lawrence et al. 2019), of which the performance has already been evaluated in various studies with observational data. The latest version, CLM5, is especially of interest because it includes various improvements over previous versions. For example, Kennedy et al. (2019) implemented a new plant hydraulic stress parameterization and showed

improvements in simulating transpiration and soil water content of a tropical forest site. In addition, Swenson et al. (2019) improved CLM5 further by implementing lateral flow, i.e., water fluxes within a CLM5 grid cell between soil columns with different slopes, reproducing differences in evapotranspiration between upland and lowland hillslopes. In many studies, comparisons of CLM model results were made with in situ observations using a single grid cell setup. For example, Hudiburg et al. (2013) used a single-point setup of the CLM4.0 model to predict net and gross primary production of forested FLUXNET

sites in Oregon, USA. Similar CLM single-point setups were also used to perform model sensitivity studies. For instance, Zhang et al. (2019) adjusted vegetation phenology parameters of the temperate grassland plant functional type in CLM4.5 to reduce an overestimation of growing-season LAI and annual gross primary production, while enhancing the partitioning of evapotranspiration for the study site. Similarly, Post et al. (2017) also used CLM4.5 single-point setups to estimate net carbon fluxes at four European sites and they improved the assessment of annual net ecosystem exchange by estimating ecosystem

parameters using a Markov chain Monte Carlo method. Wieder et al. (2017) used CLM4.5 to investigate the impact of extending growing seasons on carbon, water, and energy fluxes and found that of the five ecosystems considered, wetland ecosystems were the most affected.

On the other hand, observational SWC data also face various limitations and uncertainties (Vereecken et al., 2008). For instance, high-quality in-situ SWC measurements usually only cover relatively small areas, while remote sensing observations

give only indirect information about SWC for the upper few centimeters of the soil at relatively coarse spatiotemporal resolution. Nevertheless, a growing number of soil moisture products from remote sensing has become available e.g. Soil Moisture and Ocean Salinity (SMOS) (Kerr et al., 2010), Soil Moisture Active Passive (SMAP) (Entekhabi et al., 2010), European Space Agency Climate Change Initiative (ESA-CCI) (Dorigo et al., 2017), which are used to improve the accuracy of land surface model predictions, e.g. of soil moisture, energy and carbon fluxes, through data assimilation.


Data assimilation aims at optimally merging model simulations and measurement data, according to statistical optimality principles, so that the uncertainty of the model simulations is reduced and the accuracy improved. It is common practice that numerical models are implemented without intrinsic data assimilation and external frameworks are used to perform data assimilation. Coupling to such external codes instead of implementing data assimilation inside the numerical model provides
many advantages. External frameworks are usually built for modularity and extendibility, i.e., they provide multiple different data assimilation methods and can be updated when new methods are developed. Additionally, external frameworks are usually optimized for parallel computing. We can distinguish between two different approaches for the coupling of models with external frameworks. In case of offline coupling, the framework wraps around the model and does not modify the model source code but instead interfaces with the model through output files. This non-intrusive method uses the input, output and restart
functionalities of the model to perform data assimilation. In contrast, the online coupling framework is incorporated into the model code, which allows to perform data assimilation in the main memory during simulation avoiding costly file input/output operations. The Data Assimilation Research Testbed (DART) (Anderson et al., 2009), which was originally developed for data assimilation with atmospheric models, is commonly used for offline coupled data assimilation with all components of the Earth system within the Community Earth System Model, including land, atmosphere, ocean, sea/land ice, and other earth system
models. While the studies cited in this section use DART for offline coupled data assimilation, we were made aware that the use of DART for online coupling is in development. Recently some studies have shown its application in combination with CLM. For example, Zhang et al. (2014) assimilated satellite snow cover fraction data from MODIS (Moderate Resolution Imaging Spectroradiometer) into CLM4.0 using DART, which led to improved snow depth predictions. Ling et al. (2019) assimilated the Global Land Surface Satellite (GLASS) leaf area index (LAI) product into CLM4.0 using DART. They showed
that updating both model LAI and leaf C/N can reduce the largest bias from 5m²/m² by 1m²/m² and significantly improve LAI predictions especially in forested regions. In another study, LAI and biomass observations were assimilated into a single-point CLM4.5 model for a semiarid ecosystem site in central New Mexico, USA, which improved the simulation of the carbon cycle (Fox et al. 2018). Recently, DART has also been used to assimilate brightness temperature data from the Advanced Microwave Scanning Radiometer for Earth Observing System (AMSR-E) into CLM4.0 on a global scale to improve the prediction of soil
water content (Zhao et al., 2016). In this study, it could be shown that soil water content simulation can be improved by data assimilation, but some of the systematic biases of CLM4 simulations could not be resolved. Raczka et al. (2021) used DART to assimilate remotely sensed leaf area index and above ground biomass in CLM5 to improve carbon flux simulation . The Parallel Data Assimilation Framework (PDAF) (Nerger et al. 2005) has also been used in various studies to assimilate SWC measurements into different CLM model versions. For example, Shrestha et al. (2018) successfully used PDAF to perform
joint state and parameter updates with CLM3.5 to improve soil moisture prediction and suggested that this approach is applicable to CLM5. In a more recent study, PDAF was used to assimilate the ESA CCI microwave soil water content product in CLM3.5 with the ensemble Kalman filter to improve European predictions of soil water content and runoff estimations (Naz et al. 2019, 2020).

In this study, we present the coupling of PDAF as a framework for the data assimilation because it provides many data

assimilation algorithms, supports online coupling, and includes templates for the modifications to the model code that are necessary for the coupling with CLM5. In general, online coupling is important in high performance computing to avoid time consuming file read/write operations. In this regard, Nerger et al. (2013) and Kurtz et al. (2016) have demonstrated the excellent scaling and performance of PDAF, for which reason we selected PDAF for our data assimilation study with CLM5. Additionally, PDAF is also part of the modular Terrestrial System Modelling Platform (TSMP) (Shrestha et al. 2014). PDAF

has previously been coupled to CLM 3.5 within TSMP (Kurtz et al. 2016) and thus coupling PDAF to CLM5 has the potential benefit of simplifying future couplings to the other components of TSMP. The new developments in this study for integrating CLM5 into the TSMP environment include changes to the interface to CLM5 and a new software environment, which are described in detail in Section 2.3.

To illustrate the potential of the CLM5-PDAF coupling, we also present an application using the ensemble Kalman Filter to

perform simultaneous state and parameter updates in the Wüstebach forest headwater catchment. The Wüstebach catchment is part of the TERENO network and various hydrological models have already been applied to it, e.g. HydroGeoSphere (Cornelissen et al., 2016; 2014); MIKE-SHE (Koch et al., 2016) and CLM-Parflow (Fang et al., 2015; 2016). Some of these modelling studies have focused on the spatial and temporal analysis of the effect of different parameterization approaches to represent the heterogeneous soil properties (Cornelissen et al., 2014; Fang et al. 2015; 2016). Koch et al. (2016) compared

CLM-Parflow, HydroGeoSphere and MIKE-SHE and concluded that the consideration of heterogeneous porosities can increase model performance depending on the model structure. However, in earth system modelling applications, distributed simulation of such small catchments is usually computationally not feasible and a single grid cell is used instead. With such coarse scale applications in mind, and to demonstrate the application of CLM5-PDAF in a simplified model setup, we represent the Wüstebach catchment by a single grid cell. Furthermore, using a single grid cell approach, we can highlight the

improvements data assimilation and parameter updating can provide for correcting biases in the system and errors in the parameters.

In this paper, we present the development of the coupling of the latest version of CLM (CLM5) with PDAF and explore the potential of data assimilation in CLM5 and its potential for updating model parameters. Furthermore, we investigate whether updating of the soil organic matter parameter via data assimilation can further improve the prediction of soil water with CLM5.

The paper is structured as follows: First, we give a short description of CLM5 and PDAF and then explain in detail how their coupling was realized. We then present the study site, the data used for the simulations, and the results for different data assimilation scenarios. We end with a discussion, conclusions, and an outlook on further planned improvements, for example concerning parameter updating.

## 2 Methods

### 2.1 Model description

In this study, the Community Land Model 5.0 (CLM5) (Lawrence et al. 2019) is used to simulate land surface processes, in particular hydrological processes such as infiltration, evaporation from both soil and vegetation, transpiration, surface runoff and sub-surface drainage. The new plant hydraulic stress parameterization by Kennedy et al. (2019) impacts both the soil water content and also the coupling to the carbon cycle. We focus in particular on the simulation of the distribution and temporal dynamics of soil water within the soil column. Surface runoff is simulated in CLM5 using the SIMTOP model (Niu et al. 2005), which is based on the TOPMODEL approach (Beven and Kirkby 1979). Compared to previous versions, CLM5 allows a spatially variable soil depth with an underlying, impermeable bedrock. This replaces the unconfined aquifer parameterization (Niu et al. 2007) of previous versions with a zero flux lower boundary condition and an explicit water table depth (Lawrence et al. 2018). Sub-surface drainage is calculated as a function of an ice impedance factor, a base flow calibration parameter, the topographic slope, and the thickness of the saturated part of the soil column (Lawrence et al. 2018). The distribution and temporal evolution of soil water within the soil column is calculated with a finite-difference approximation of the Richard's equation including Brooks-Corey parameterization. The hydraulic parameters involved in these calculations are determined by a weighted combination of mineral and organic properties. The mineral component of the soil hydraulic parameters is determined by pedotransfer functions and depends on sand and clay fractions (Clapp and Hornberger 1978). Starting with the version 4 of CLM the hydraulic parameters are also depending on organic matter content in the soil. For example, without the contribution of organic matter, the soil porosity in CLM is limited to a maximum of 0.489 for soils without sand fraction due to the implemented pedotransfer function of Clapp and Hornberger (1978). However, as can be seen in Figure 3, the soil water content observations in the Wüstebach catchment show frequently higher values. Incorporating the new equations with soil organic matter content increased the maximum value for porosity at the surface to 0.93 with decreasing porosity values with increasing soil depth. This shows that in order to simulate soil moisture in forest soils with high porosity, it is important to consider organic matter. The detailed equations for accounting for organic matter on soil hydraulic parameters can be found in Appendix A.

The numerical solution of the Richard's equation in CLM5 is based on a linearization that leads to a tridiagonal system of equations (Lawrence et al. 2018). CLM5 uses an adaptive time-stepping solver (Clark & Kavetski 2010, Kavetski et al. 2001) that improves the numerical stability for frozen soils and shallow bedrock compared to solvers in previous versions.

### 2.2 Data assimilation framework

#### 2.2.1 Ensemble Kalman Filter

In Earth sciences, two common data assimilation approaches are 1) variational methods, often used in atmospheric models, and 2) sequential methods like the Ensemble Kalman filter (Reichle 2008). The Kalman Filter originates in filtering and prediction of linear dynamic systems (Kalman 1960) and the Ensemble Kalman Filter (EnKF) is a stochastic approximation

for nonlinear dynamic systems based on Monte Carlo methods (Evensen 1994, Burgers et al. 1998). Included in PDAF are implementations of the most common variants of the Kalman filter. This study uses exclusively the ensemble Kalman filter (EnKF), in which an ensemble of independent model simulations is used to approximate the model error covariance matrix from the spread of the ensemble. For nonlinear models, like CLM5, ensemble spread is created from perturbations of model

parameters and model forcings individually for each ensemble member. During the simulations, the EnKF uses an update step to assimilate observational data at time steps where observations are available. The update step is described by the following equation:

$$\mathbf{x}_a^i = \mathbf{x}_f^i + \mathbf{K}[\mathbf{y} - \mathbf{H}\mathbf{x}_f^i] \ (1)$$

where the superscript $i$ refers to ensemble member $i$, $x_a^i$ is the updated state vector after the analysis, $x_f^i$ is the forecasted model

state vector, $\mathbf{K}$ is the Kalman gain, $\mathbf{y}$ is the observation vector, and $\mathbf{H}$ is the so-called measurement operator that transforms between model and observational states. Observational data is perturbed for each ensemble member to maintain the correct error statistics (Burgers et al. 1998). Therefore, y in equation 1 is shorthand for y=o+i, where o is the observational data and i is a perturbation vector with mean zero and covariance according to the observational error covariance matrix. For simplicity, the observational error is assumed to be constant and set to a root mean square of 2%. The Kalman gain $\mathbf{K}$ represents the

weighting of observations versus model and is computed as follows:

$$\mathbf{K} = \mathbf{P}\mathbf{H}^T \ (\mathbf{R} + \mathbf{H}\mathbf{P}\mathbf{H}^T)^{-1} \ (2)$$

where the superscript T refers to transposed matrices, $\mathbf{P}$ is the model error covariance matrix and $\mathbf{R}$ is the observational error covariance matrix. Therefore, the Kalman gain represents how much the model error contributes to the total error. Conceptually, $\mathbf{K}$ approaches 1 if the observational error covariance is very small compared to the model error covariance which in Eq. 1 would result in more weight for the correction based on the observational data. On the other hand, $\mathbf{K}$ approaches

0 if the observational error covariance is much larger than the model error covariance resulting in a smaller weight for the update term in Eq. 1. The observational error covariance matrix $\mathbf{R}$ is often statistically defined based on the measurement error of the observations which are usually assumed to be independent. The model error covariance matrix $\mathbf{P}$ in the Ensemble Kalman Filter is approximated using the ensemble statistics. Specifically,

$$\mathbf{P} = \frac{1}{(N-1)} \sum_{i=1}^{N} (\mathbf{x}_f^i - \overline{\mathbf{x}_f})(\mathbf{x}_f^i - \overline{\mathbf{x}_f})^T \ (3)$$

where N is the number of ensemble members and $\bar{x}$ is the ensemble mean. For example, ensemble members can be generated based on perturbed soil parameters and atmospheric forcings. The perturbations of soil properties and forcings represent the uncertainty range of the model, the specifics of the ensemble generation for this study are described in Sections 3.2.2 and 3.2.3.

Only during the data assimilation update step the ensemble members are connected through Eq. 3. Therefore, the ensemble Kalman filter is well-suited for parallelization. See Kurtz et al. (2016) for a discussion of the scaling of the Ensemble Kalman filter in PDAF. Each ensemble member is independently propagated in time.

In this study, the observation vector y contains the soil water content observations, described in Section 3.2. The state vector $x^i$ contains soil water content (model states), sand and clay fractions (parameters), and organic matter fractions (parameters) depending on the experiment as described in Section 3.3. CLM5 uses a subgrid hierarchy that contains land units, columns, and patches. Patches represent different plant functional types and share a single column. The physical state variables, like soil water content, are defined at column level and vertically discretized into layers. There are up to 20 hydrologically active layers depending on the depth to bedrock parameter. For simplicity, we consider the model state for soil water content to be the 20 layers of the column even if not all 20 layers are active. Specifically, we use the diagnostic soil water content variable called "H2OSOI" as the model state variable and after each update adjust the prognostic liquid and solid water content variables "H2OSOI_LIQ" and "H2OSOI_ICE". The measurement operator in this case is a simple mapping of the three observation vector components to the state vector component at the corresponding depth.

### 2.2.2 Parameter updating

In this study, we also apply a joint state and parameter estimation approach to further improve simulation results. Specifically, the state augmentation approach (Friedland, 1969; Fertig et al., 2009) is applied in which the forecasted model state vector ($x_f^i$ in Eq. 1) contains both the model state variables and relevant model parameters. The attached model state parameters are updated based on the Kalman gain (Eq. 2) without direct observations of the model parameters.

For assimilation of soil water content the relevant model parameters are the hydraulic parameters. A common approach is indirect updating the hydraulic parameters by updating the soil texture, i.e. sand and clay fraction, and using the pedotransfer function as described in Section 2.1 (e.g. Naz et al., 2019). A more consistent approach would be to update the hydraulic parameters directly instead of updating the soil characteristics and using the pedotransfer function. However, since the existing implementations use the indirect approach we chose to follow the same approach in this study. Before CLM version 4.0, only sand and clay fractions were used to calculate the hydraulic parameters and therefore previous couplings of CLM and PDAF did not include organic matter as an option for joint state and parameter estimation. Similar to the work of Han et al. (2014) for CLM 4.5, we added organic matter as an additional parameter, which can be updated with the CLM5-PDAF coupled model.

### 2.3 Coupling CLM5 with PDAF

As previously mentioned, this study makes use of the highly modular nature of TSMP (Shrestha et al. 2014) to integrate CLM5 as a new option for the land surface model component in the coupling framework. TSMP is designed to couple combinations

of an atmospheric model, e.g. COSMO (Baldauf et al. 2011), a land surface model, e.g. CLM (Oleson et al. 2008), a sub-
surface model, e.g. ParFlow (Ashby and Falgout 1996; Kollet and Maxwell 2006), and a data assimilation framework, e.g.
PDAF (Nerger et al. 2005). The modularity allows not only the realization of a fully coupled system of all components, but
also combinations like CLM and ParFlow or CLM and PDAF and also individual model components can be executed.

This study focuses on the implementation of the coupling of CLM5 and PDAF inside the TSMP framework. However, an
advantage of implementing this single pair coupling inside a larger, modular platform is to facilitate future coupling
implementations to the other components of TSMP. In general, the coupling in TSMP uses the Ocean-Atmosphere-Sea-Ice-
Soil coupler – Model Coupling Toolkit (OASIS-MCT) (Valcke et al., 2013) to couple the models in a multiple program
multiple data (MPMD) approach. However, as described in Kurtz et al. (2016), coupling with PDAF is an exception to this
approach. Instead of using MPMD, a single executable is built out of modified, pseudo-library versions of the models. This
keeps all model data in main memory and avoids I/O intensive re-initialization of models. Additionally, since in this study
only one model (CLM5) and PDAF are coupled, the utilization of OASIS-MCT is not necessary.

Figure 1 sketches the organization of the CLM5-PDAF coupling into five main components. The next paragraphs describe
these components in more detail and modifications compared to the CLM3.5-PDAF implementation by Kurtz et al. (2016) are
discussed. The PDAF components, core functions and user functions, are the same as described in Nerger et al. (2005) and
Kurtz et al. (2016) respectively. The new code developments in the PDAF user functions are superficial inclusions of CLM5
as option with the same functionality as already implemented and described by Kurtz et al. (2016) for CLM 3.5.

The main program, labelled TSMP-PDAF driver, controls the individual components and handles the parallel communication
using multiple MPI communicators. Adding CLM5 coupling requires only minor changes to the TSMP-PDAF driver to add
CLM5 as a new option to the models controlled by the driver.

The TSMP wrapper contains the majority of additional code for coupling CLM5 and PDAF. The TSMP-PDAF driver uses the
TSMP wrapper as an interface to the individual pseudo-libraries of the models. Therefore, the TSMP wrapper contains the
modified routines from the model for initialization, time stepping, and clean-up. The development of CLM5-PDAF includes
modifying these routines from the original CLM5 source code. These routines are moved from the CLM5 default driver, which
is taken from the Common Infrastructure for Modelling the Earth (CIME) framework, into the TSMP wrapper. The clean-up
routine is migrated without modification. The modification to the initialization routine involves an added call to the subroutine
that defines the state vector. The main time stepping loop in CLM5 works by looping until a stop alarm is received. On the
other hand, the TSMP framework, similar to older versions of CLM, works with a loop counting up until a specified end time
is reached enabling data assimilation at specified time steps. Therefore, the TSMP wrapper subroutine to advance CLM5
contains only the code from inside the original time stepping loop. In this way, the TSMP-PDAF driver can control how many
CLM5 time steps are performed before stopping for an interrupting data assimilation step. Further modifications to the time

stepping subroutine include the addition of calling the PDAF specific subroutine to set the state vector before each data assimilation step.

Additionally, the TSMP wrapper contains the model specific routines for managing the PDAF state vector. As these routines are model dependent, part of the development of CLM5-PDAF included the creation of routines to interface with CLM5. This includes defining the size of the state vector based on domain decomposition, for non-single grid cell simulations and options for parameter updating. The TSMP wrapper provides both the subroutine called by the model to set the state vector and the subroutine called by the data assimilation method to update the model variables contained in the state vector. For soil water content and soil texture parameters setting the state vector is simply copying the model values to their respective place in the state vector. The subroutine to update the state vector contains functionality to detect and correct invalid values, e.g. below residual soil water content, above porosity, and below 0% or above 100% for the sum of the sand and clay fractions. Furthermore, for the optional parameter updating it is necessary to provide a function to transform the input parameters, e.g. soil texture, to the model parameters, e.g. the soil hydraulic parameters. CLM5 performs this transformation once during initialization to obtain the hydraulic parameters from the soil texture in the surface file. As mentioned in Section 2.1, this procedure has changed compared to older versions of CLM. The subroutine to perform this transform after each data assimilation step follows the implementation in CLM5 and is shown in Appendix A.

The component labelled libclm5 in Figure 1 is the pseudo-library from CLM5 compiled modules. Code modifications for CLM5 source files are limited to two driver modules related to parallel communication and ensemble reading of name list files. As previously mentioned, the TSMP-PDAF driver manages the initialization of the parallel communication that involves initializing MPI and splitting the global communicator MPI_COMM_WORLD into specific model, filter, and coupling communicators. However, by default CLM5 also initializes MPI and uses MPI_COMM_WORLD for its parallel communication. Since only one MPI_COMM_WORLD can exist within a MPI application, the CLM5 code was modified to not initialize MPI and not use MPI_COMM_WORLD.

In ensemble simulations each member has individual input files. In CLM input files are controlled by name lists. In older versions of CLM a single name list was used and to enable ensemble simulations for TSMP-PDAF only involved attaching an ensemble identifier suffix to the name of this name list. In CLM5 there are multiple name lists and managing the reading of them has become more complex. However, CLM5 also supports an ensemble mode where each ensemble member reads name lists with identifier suffixes. Our implementation of CLM5-PDAF makes use of this ensemble mode. The ensemble mode is modified such that it uses the PDAF model communicator instead of splitting the global communicator. Therefore, the initialization subroutine that handles the ensemble mode is modified to accept a communicator and an individual ensemble member number from PDAF. Additionally, the initialization subroutine also passes the PDAF information to the subroutine that initializes the communicators for CLM5 and replaces the default ensemble mode identifiers with the PDAF specific identifiers. Figure 2 illustrates these modifications and shows the general process flow difference between CLM5 and CLM5-PDAF, i.e., the interruption of the CLM simulation by the PDAF data assimilation step.

## 3. Test case

### 3.1 Study Site

The coupled modelling framework is applied to the small (38.5 ha) forested catchment called Wüstebach which is located in the Eifel National Park near the Belgium-Germany border. As part of the Terrestrial Environmental Observatories (TERENO) network (Bogena et al., 2015; Bogena et al., 2018), the Wüstebach site has a wireless sensor network (SoilNet) to provide soil water content and soil temperature measurements since 2009 at 5, 20, and 50 cm depth at 150 locations every 15 minutes (Bogena et al. 2010).

The Wüstebach test site is also interesting because in the late summer / early autumn of 2013 the national park forest management removed the prevailing spruce monoculture forest in an area to promote the natural regeneration of deciduous forest. The SoilNet was installed before this change, so that the impact on the soil water content is measured before and after this land use change. However, in this study we use the study site mainly to demonstrate the functionality of the newly coupled CLM5-PDAF framework and therefore, we focus on the undisturbed forested area.

As mentioned in the introduction, we do not focus on spatial heterogeneity but instead look at the study site as it would be modelled in a regional or continental simulation, i.e., as a single grid cell. This allows for a clear and simple setup to test and demonstrate the functionality of CLM5-PDAF and simultaneously allows us to use a larger ensemble than is usually feasible for regional or continental data assimilation simulations.

For the modelling, we use a grid cell size of 3 by 3 km based on the grid used in Naz et al. (2019) for a continental scale study. Unless specified, we used the default parameters of CLM5 and followed the instructions of the online CLM5 user guide to get initial soil characteristics, topography, and other initial parameters of the surface file. The model was spun-up from a cold start as described in the CLM5 user guide with atmospheric forcings from 2009 to 2018 described in more detail in Section 3.2.2. More specific details on the different simulation setups are presented in Section 3.3.

### 3.2 Data

#### 3.2.1 Soil water content – in-situ measurements

The observational data of the study site Wüstebach is pre-processed before assimilation. The raw data from the TERENO data portal (Sorg et al. 2015) contains data for all stations and all sensors in 15 minutes intervals including quality flags. The observational data is the soil water content, i.e. the ratio of the volume of water to the porosity. This data is pre-processed using filters that remove data points based on their quality flag, spikes, frozen soil condition, and erroneous values. Spikes are defined as reductions in soil water content of more than 1% or increases in soil water content of more than 5% with an

immediate return to values within 1% of the value before the spike. Soil water content below 1% or above 90% is considered erroneous. These thresholds and the definition of spikes are based on Wiekenkamp et al. (2016) and Dorigo et al. (2013). In Wüstebach each soil water content sensor is paired with a soil temperature sensor. This allows for the removal of unreliable measurements due to frozen soil. Time steps in which less than 25% of all sensors provide data are filtered out. The filtered raw data is then spatially and temporally averaged to fit our setup of the model, i.e., daily averages for the three soil depths from the average of the selected stations.

As mentioned above, the Wüstebach was partially deforested in 2013, with SoilNet SWC sensors covering both the undisturbed and deforested areas. The deforested part of the Wüstebach catchment is mainly located in the riparian zone featuring shallow groundwater that is strongly influenced by incoming lateral flows within the catchment. However, lateral flows are only represented through routing to rivers in CLM5. Therefore, we omitted the riparian zone and selected only SoilNet stations located in the groundwater distant forested parts of the Wüstebach catchment in this study. With these criteria 37 soil water stations remain in the forested part of the Wüstebach catchment and are used in this study.

### 3.2.2 Atmospheric forcings

The atmospheric forcings used in this study are measurements of air pressure, shortwave radiation, relative humidity, 2m air temperature, and wind speed from an on-site meteorological station. Additionally, the precipitation data is provided by the meteorological station Kaltenherberg (DWD, German Weather Service) located 5km west of the Wüstebach study site (Bogena et al., 2015). The atmospheric forcing data is perturbed to generate an ensemble for data assimilation using the EnKF. In this study, the perturbed variables are precipitation, shortwave radiation, longwave radiation, and air temperature. These variables are perturbed according to cross-correlation coefficients derived from global observations by Reichle et al. (2007). The specific perturbation characteristics used in this study are from Han et al. (2014) and shown in Table 1.

### 3.2.3 Surface parameters

The over 70 different surface parameters included in each CLM5 surface file are generated by the tools provided by CLM5 from remapping of various pre-processed global files, see Lawrence et al. (2019) for details. For the single grid cell of the study site, all default values were used, except for the plant functional type and the depth to bedrock. We chose the plant functional type "needle leaf evergreen temperate tree" to represent the spruce monoculture of the Wüstebach site. The depth to bedrock was adjusted to 1.6 meters according to Fang et al. (2015). Sand, clay, and organic matter fractions are perturbed for each ensemble member. Perturbed values were obtained by drawing from a uniform distribution with mean zero and a range between -20% and +20%. Perturbations that cause the sum of sand and clay fractions to exceed 100% are re-scaled to be limited to 100%. These perturbations are larger than, for example, the ones used in Han et al. (2014) to represent a larger initial model parameter uncertainty for a single grid cell simulation with a larger ensemble.

## 3.3 Simulation experiments

Four different setups were used to demonstrate the functionality and effectiveness of CLM5-PDAF. The open loop (OL) setup
has forward simulations without data assimilation. These simulations are equivalent to CLM5 standalone ensemble simulations
with perturbed inputs for both atmospheric forcings and soil characteristics. The perturbed inputs represent both forcing and
model uncertainty and determine the ensemble variance. The initial data assimilation setup limits the state vector to the soil
water content variable (DA_s). The data assimilation with state and parameter updates setup (DA_s+p) applies the joint state
and parameter estimation approach, described in Section 2.2.2, by augmenting the state vector with sand and clay fractions.
The fourth setup, data assimilation with state and parameter updates including organic matter (DA_s+p+o) adds the soil organic
matter fraction to the state vector. All setups were run for a 10-year period starting from 2009 when observations become
available.

We used four statistical metrics to evaluate the quality of the simulation results: the root-mean-square-error (RMSE), the
unbiased root-mean-square-error (ubRMSE), the mean bias error (MBE) and the squared correlation coefficient (R²):

$$\text{RMSE} = \sqrt{\frac{\sum_{i=1}^{N}(\mathbf{Hx}^i - \mathbf{y}^i)^2}{N}} \quad (4)$$

$$\text{ubRMSE} = \sqrt{\frac{\sum_{i=1}^{N}[\left(\mathbf{Hx}^i - \overline{\mathbf{Hx}^i}\right) - \left(\mathbf{y}^i - \overline{\mathbf{y}^i}\right)]^2}{N}} \quad (5)$$

$$\text{MBE} = \frac{\sum_{i=1}^{N}(\mathbf{Hx}^i - \mathbf{y}^i)}{N} \quad (6)$$

$$R^2 = 1 - \frac{\sum_{i=1}^{N}\left(\mathbf{y}^i - \mathbf{Hx}^i\right)^2}{\sum_{i=1}^{N}\left(\mathbf{y}^i - \overline{\mathbf{y}^i}\right)^2} \quad (7)$$

where **y** represents observations, **Hx** represents simulated values, i is the ensemble member, N the total number of ensemble
members and overbar represents ensemble average.

## 3.4 Comparison of the four different simulation setups

Figure 3 shows time series of the monthly averaged SWC at the three observation depths. The monthly averages highlight
better the tendencies for the different simulation setups. The simulation results of all model setups show a good agreement
with the soil water content observations at 20 cm depth, while there are clear deviations at 5 and 50 cm depths. The simulations
tend to have a wet SWC bias compared to the observations at 5 and 50 cm depths but underestimate SWC at 20 cm depth. This
behaviour could be the result of the root profile in CLM5 or other uncertainties related to model parameters. This difference
is more distinct in the summer months at 5 cm depth, especially during the dry summer of 2018. The soil water content

overestimation by the model at 50 cm depth is smaller, but more consistent over time. The results of the simulations at 5 cm
depth illustrate the effects of data assimilation: All three data assimilation setups provide soil water content predictions that
are closer to the observations compared to the open loop setup simulation.

The scatter plots in Figures 4, 5, and 6 show the comparisons of the different data assimilation scenarios in terms of correlation
of daily soil water content averages between observations and simulations. Table 2 summarizes the complementary statistical
results. The evaluation at 5 cm depth, shown in the top left of Figure 4, reflects the overestimation of soil moisture content by
the open loop simulation. All observed daily average SWC below 40% are overestimated by the model. The other three scatter
plots in Figure 4 highlight the progressive effectiveness of the three data assimilation setups. While the DA_s setup still shows
overestimation of SWC compared to observations it reduces the RMSE compared to the OL setup by 30%, the ubRMSE by
35%, and increases the R² to above 0.9. The DA_s+p and DA_s+p+o setups show similar, improved results. DA_s+p performs
slightly better in terms of ubRMSE and R² than the DA_s+p+o but slightly worse in terms of RMSE and MBE.

The results at 20 cm depth (Fig. 5) show a closer agreement between observations and simulations than the results at 5cm and
50cm depth. At 20cm depth, simulations slightly underestimate SWC. Similar to 5cm depth, the DA_s improves the RMSE
by 30% compared to OL and increases the R² to above 0.9. At 20 cm depth, the DA_s+p+o shows an especially small MBE
and overall very good agreement with the observations, suggesting that updating the organic matter faction does contribute to
more accurate simulation results.

The results from 50 cm depth (Fig. 6) show the most consistent overestimation of SWC by the model and the smallest
improvement by data assimilation. The DA_ssetup reduces the RMSE by only 7% compared to the OL and even the best
performing setup (DA_s+p+o) only improves RMSE by 15%. The DA_s+p and DA_s+p+o scenarios result in similar results
at 5cm depth (Fig. 4).

Figure 7 shows the impact of the soil water content data assimilation on the evapotranspiration flux (ET). For all statistical
characteristics, the impact is negligible for the three data assimilation scenarios. We believe this is due to the overall wetness
of the study area, i.e., the soil water content is not the limiting factor for ET, so other variables or parameters would need to
be assimilated to affect simulated evapotranspiration.

Table 3 shows the changes in soil texture related to the parameter updates. The parameter updates increase the sand fraction
at all three measurement depths by a factor of 2. The clay fraction, on the other hand, is only slightly reduced across these
depths. Organic matter fraction is also increased in all three depths, but more significantly in 5 cm and 20cm. For this particular
study site, there are measurements for the soil characteristics at various points throughout the catchment. However, we did not
perform comparisons between the updated soil characteristics values and measurements, since it is not simple to overcome the
heterogeneity of discrete spatially distributes point measurements of soil characteristics and scientifically combine them into
a coarse catchment scale value.

## 4. Discussion and Conclusions

In this study, we presented the newly coupled data assimilation framework CLM5-PDAF. The presented implementation can be summarized by the following three main aspects, which are discussed in this section: the online variant of PDAF, re-use of CLM5 ensemble mode, and the TSMP framework. The online variant of PDAF performs data assimilation in the main memory during runtime by coupling the model and PDAF in a single executable. We have described the necessary code modifications to achieve this coupling. The presented implementation re-uses the CLM5 ensemble mode which enables multiple simulations to run in parallel from the same executable while using independent inputs and creating individual outputs. This re-use minimizes necessary code changes to connect CLM5 and PDAF and simplifies the management of the parallel communicators of CLM5 and PDAF. The framework of TSMP provided the build infrastructure and the template for the coupling components. We chose to include CLM5-PDAF in the TSMP to make it available for future developments in the modular environment and facilitate future couplings to other components. The performance of the CLM5-PDAF data assimilation system was illustrated with the assimilation of soil water content data for the Wüstebach site in Germany. Data assimilation decreases the mismatch between observations and model states. We further showed that including parameter updates could improve overall estimations, although some systematic bias remains. Updating also organic matter fraction, as one of the parameters determining the soil hydraulic properties, has an overall positive effect. However, even with this addition some significant differences between simulated and observed values remain, especially at 5cm depth and in dry years. We were not able to show significant impact of the assimilated soil water content on the evapotranspiration flux. In a future study, we will investigate whether this for other study sites, also in other climates. We will also include other variables and parameters in the data assimilation to test their effects on the evapotranspiration flux.

The performance of CLM5-PDAF could be further improved by updating soil hydraulic parameters themselves, instead of indirectly updating them via soil texture and pedotransfer functions. This could potentially reduce the model uncertainty further since the accuracy of the pedotransfer functions would be less of an issue after parameter updating. This will require more fundamental code changes and will be considered in future work. In addition, CLM5-PDAF will be further extended by the assimilation of more state variables, like for example LAI or soil temperature.

## Appendix A: CLM5 specific equations relating sand, clay, and organic matter fractions to soil hydraulic parameters

In CLM5 the soil hydraulic parameters are determined by a weighted average of the respective mineral and organic components. Specifically, for the mineral component the following approximations from Cosby et al. (1984) are used:

$$\theta_{(min,sat,i)} = 0.489 - 0.00126(\%sand)_i \quad (A1)$$

where $\theta_{(min,sat,i)}$ is the porosity of the mineral part and subscript $i$ refers to the vertical level.

$$B_{(min,i)} = 2.91 + 0.159(\%clay)_i \quad (A2)$$

where $B_{(min,i)}$ is the hydraulic conductivity exponent of the mineral part.

$$k_{(min,sat,i)} = 0.0070556\left(10^{-0.884+0.0153(\%sand)_i}\right) \text{ (A3)}$$

where $k_{(min,sat,i)}$ is the saturated hydraulic conductivity of the mineral part.

$$\Psi_{(min,sat,i)} = 10 \text{ (A4)}$$

where $\Psi_{(min,sat,i)}$ is the saturated suction / saturated soil matric potential of the mineral part and is related to the adsorptive and capillary forces within the soil matrix.

The organic component of the soil hydraulic parameters is approximated by the following equations from Lawrence and Slater (2008):

$$\theta_{(om,sat,i)} = max(0.83, 0.93 - 0.1D_i) \text{ (A5)}$$

Where $\theta_{(om,sat,i)}$ is the porosity of the organic part and

$$D_i = \frac{depth_i}{zsapric} \text{ (A6)}$$

where $depth_i$ is the depth of the vertical level and zsapric is the depth at which organic matter takes on characteristics of sapric peat.

$$B_{(om,i)} = max(12, 2.7 + 9.3D_i) \text{ (A7)}$$

Where $B_{(om,i)}$ is the hydraulic conductivity exponent for the organic part.

$$k_{(om,sat,i)} = max\left(k_{(min,sat,i)}, 0.28 - 0.2799D_i\right) \text{ (A8)}$$

where $k_{(om,sat,i)}$ is the saturated hydraulic conductivity for the organic part.

$$\Psi_{(om,sat,i)} = min(10.1, 10.3 - 0.2D_i) \text{ (A9)}$$

where $\Psi_{(om,sat,i)}$ is the saturated suction of the organic part.


**Code availability.**

The development branch of the CLM5-PDAF coupling is freely available via Zenodo, 10.5281/zenodo.5720866 or https://zenodo.org/record/5720866 .

**Data availability.**

Soil water content data from the TERENO site Wüstebach (TERENO ID: WU_B_001 to WU_B_150) are freely available via the TERENO data portal TEODOOR (http://teodoor.icg.kfa-juelich.de/).

**Author contribution.**

L.S. pre-processed the data, developed the code, designed and performed the simulations, and prepared the manuscript. H.B., H.J.H.F. and H.V. supervised the research, co-designed the experiments, and contributed to the manuscript.

**Competing interests.**


The authors declare that they have no conflict of interest.

**Acknowledgements**.

The authors gratefully acknowledge the support by the project LIFE RESILIENT FORESTS – Coupling water, fire and climate resilience with biomass production from forestry to adapt watersheds to climate change. This project is co-funded by the LIFE 465    Programme of the European Union under contract number LIFE 17 CCA/ES/000063. Furthermore, the authors gratefully acknowledge the computing time granted through JARA on the supercomputer JURECA at Forschungszentrum Jülich. This work used data provided by the Helmholtz Association and the Federal Ministry of Education and Research (BMBF) in the framework of TERENO (Terrestrial Environmental Observatories).

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

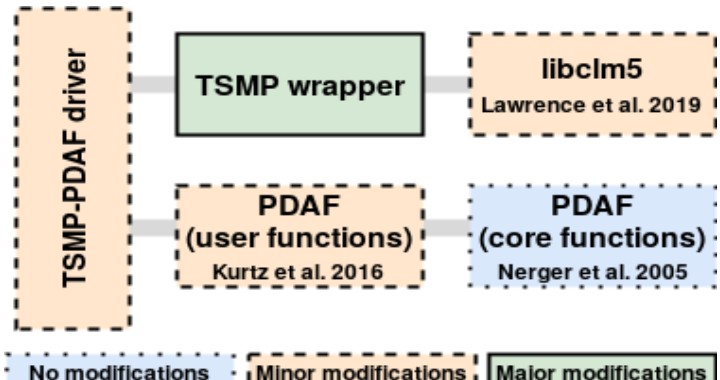

620

**Figure 1: Components of TSMP CLM5+PDAF highlighting the distinct separation of PDAF functionality, TSMP driver and wrapper, and CLM5 pseudo-library. Modifications are in relation to the implementation by Kurtz et al. (2016).**

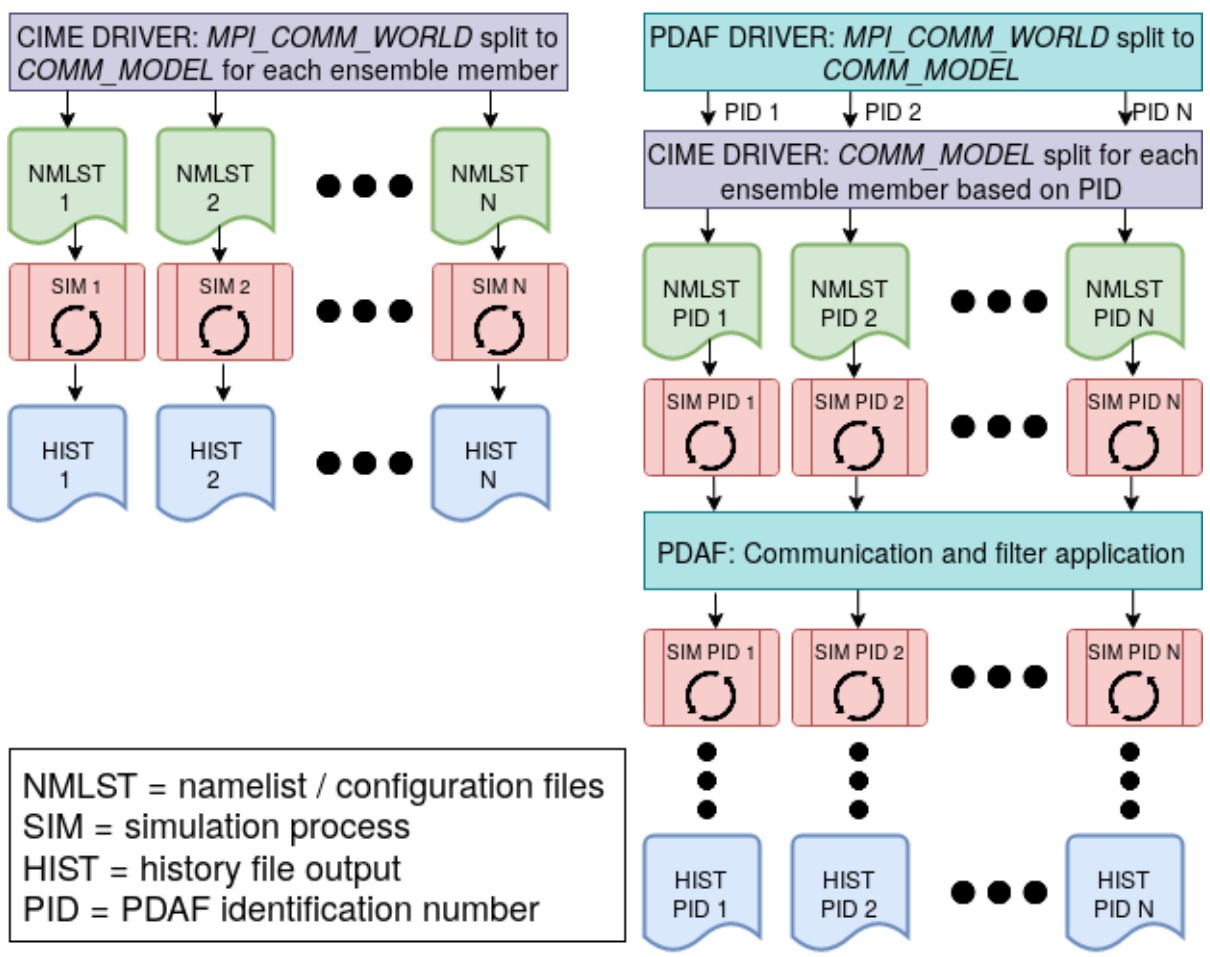

**Figure 2: Schematic overview of CLM5 ensemble mode (left side) and CLM5+PDAF (right side) communication initialization and process flow. The schema highlights the addition of communication for all ensemble members through the PDAF communication model.**

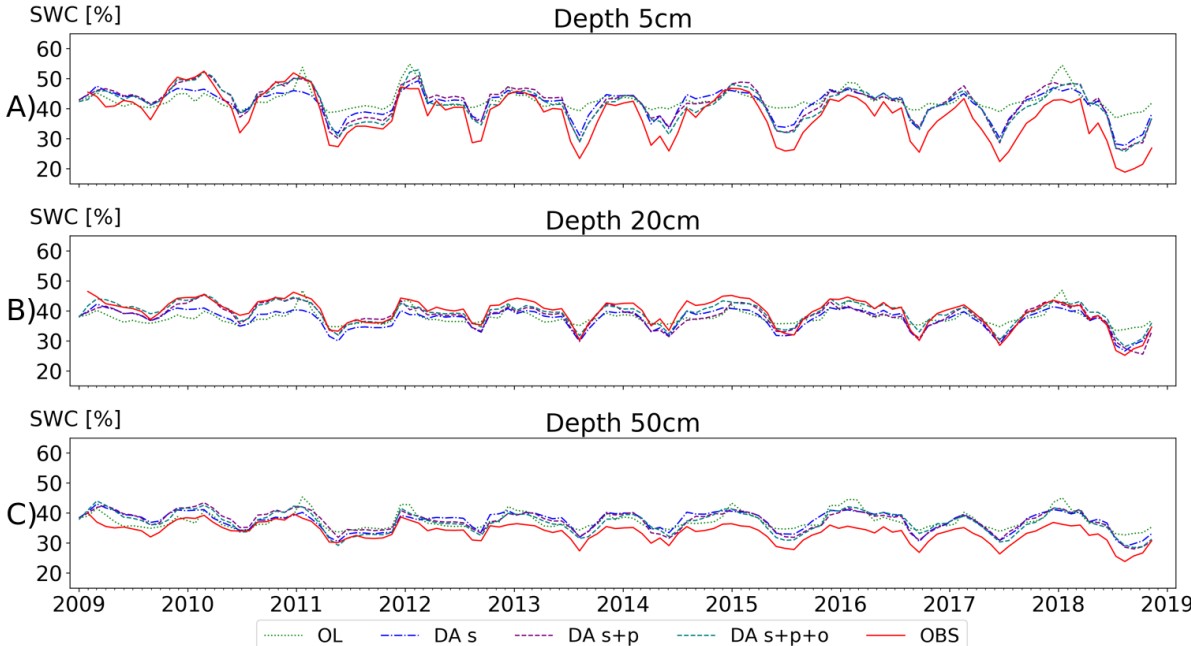

**Figure 3: Time series of the monthly averaged soil water content (SWC) from 2009 to 2018 at the three different depths and for each simulation scenario. The subplots A), B), and C) represent the three depths 5cm, 20cm, and 50cm respectively. The red (solid line) shows observational data. The light green (dotted line) shows open loop simulation results. The blue (dash-dotted line) shows results for data assimilation of state variables. The purple (dashed line) shows results for the assimilation of states and updating of parameters. The dark green (dashed line) shows results for assimilation of states and updating of parameters including organic matter.**

630

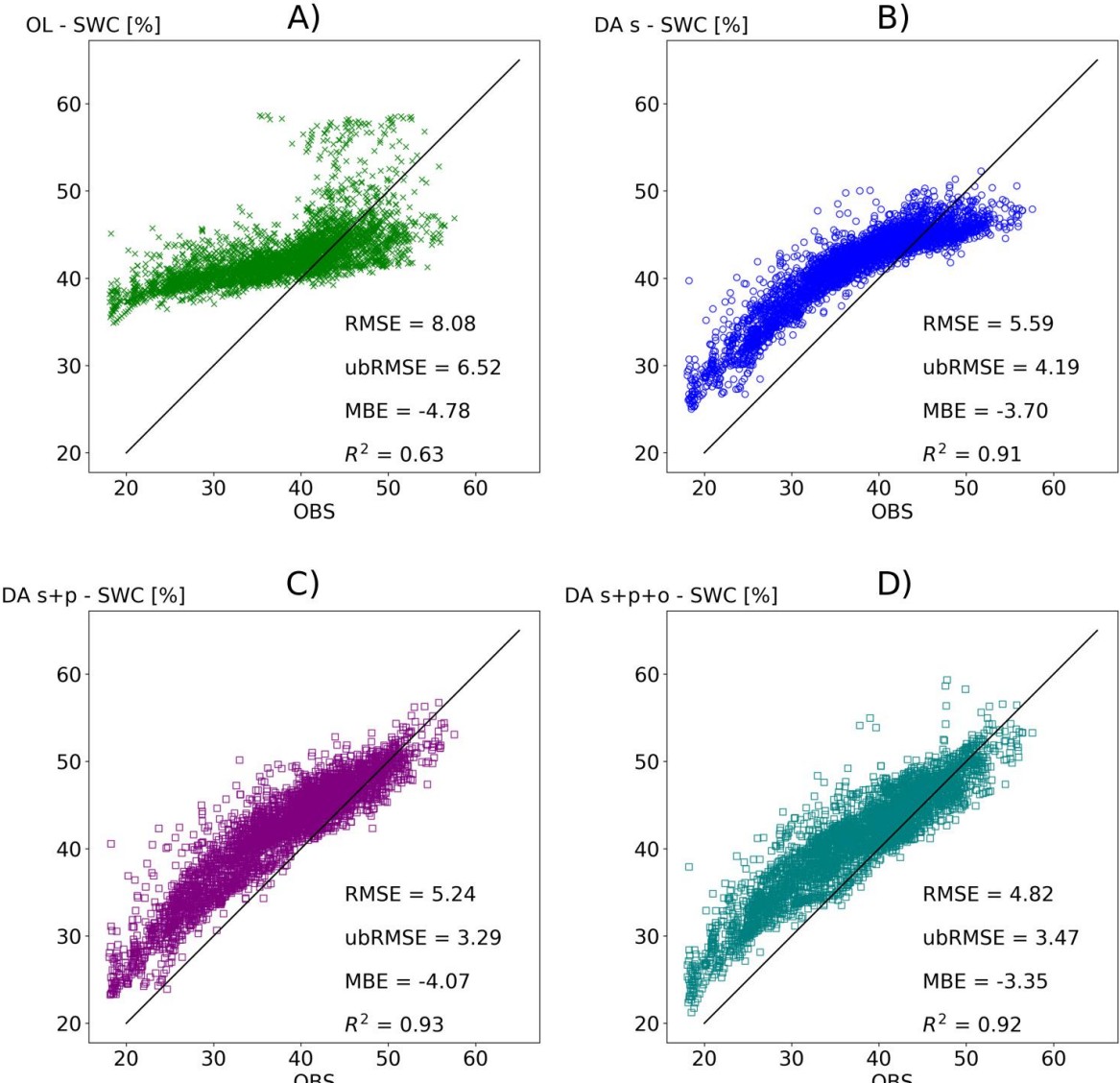

**Figure 4: Correlation diagrams for observed (OBS) and simulated soil water content (SWC) at 5 cm depth. Each marker shows one daily average. Subplot A) in the top left shows open loop (OL), subplot B) in the top right shows assimilation of state variables (DA_s), subplot C) in the bottom left shows data assimilation of state and parameters (DA_s+p), and subplot D) in the bottom right shows data assimilation of state and parameters including organic matter (DA_s+p+o). Each diagram includes the root mean square error (RMSE), unbiased root mean square error (ubRMSE), mean bias error (MBE), and squared correlation coefficient (R²).**

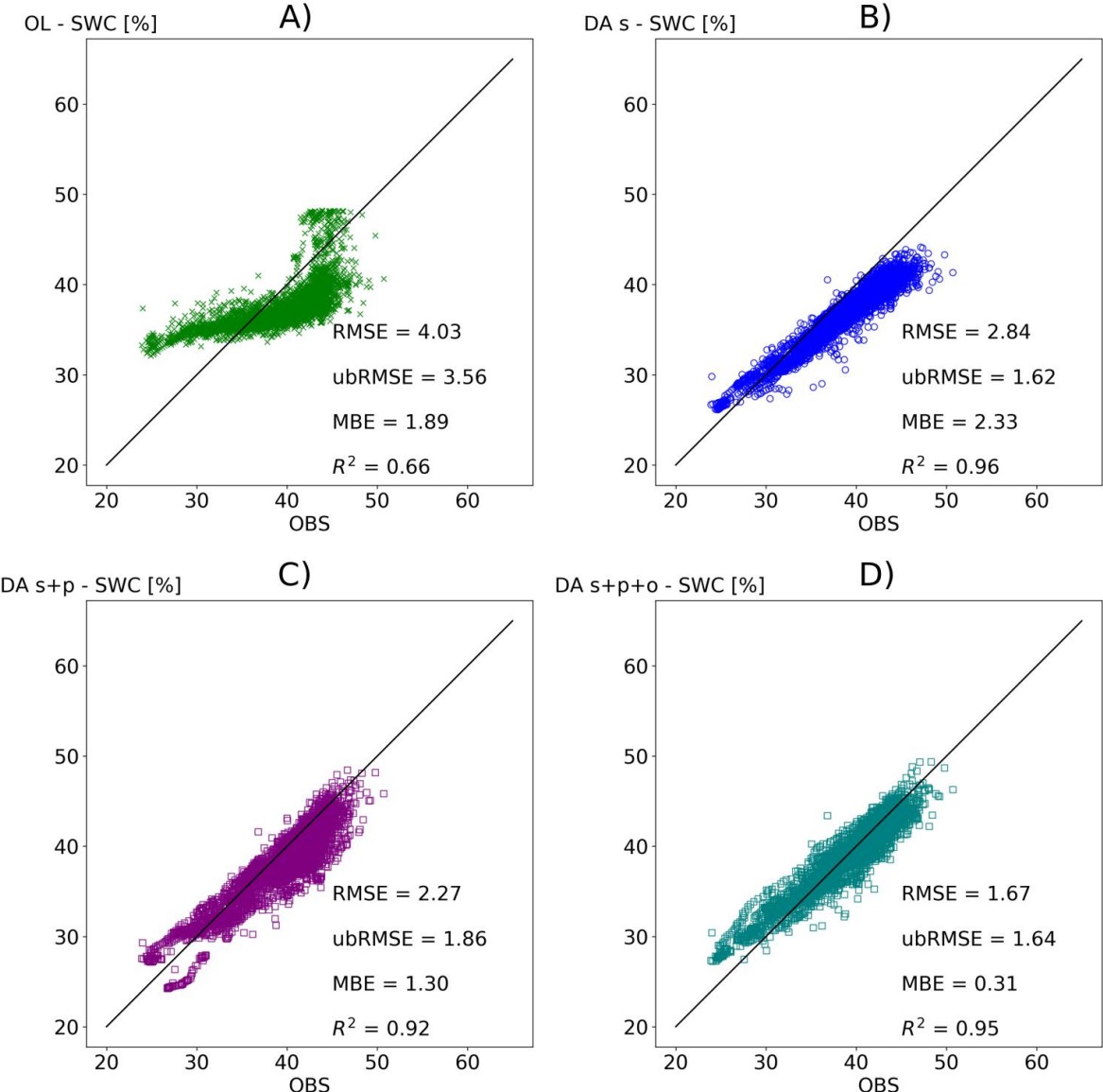

640

**Figure 5: Correlation diagrams for observed (OBS) and simulated soil water content (SWC) at 20 cm depth. Each marker shows one daily average. Subplot A) in the top left shows open loop (OL), subplot B) in the top right shows data assimilation of state variable (DA_s), subplot C) in the bottom left shows data assimilation of state and parameters (DA_s+p), and subplot D) in the bottom right shows data assimilation of state and parameters including organic matter (DA_s+p+o). Each diagram shows root mean**
645 **square error (RMSE), unbiased root mean square error (ubRMSE), mean bias error (MBE), and squared correlation coefficient ($R^2$).**

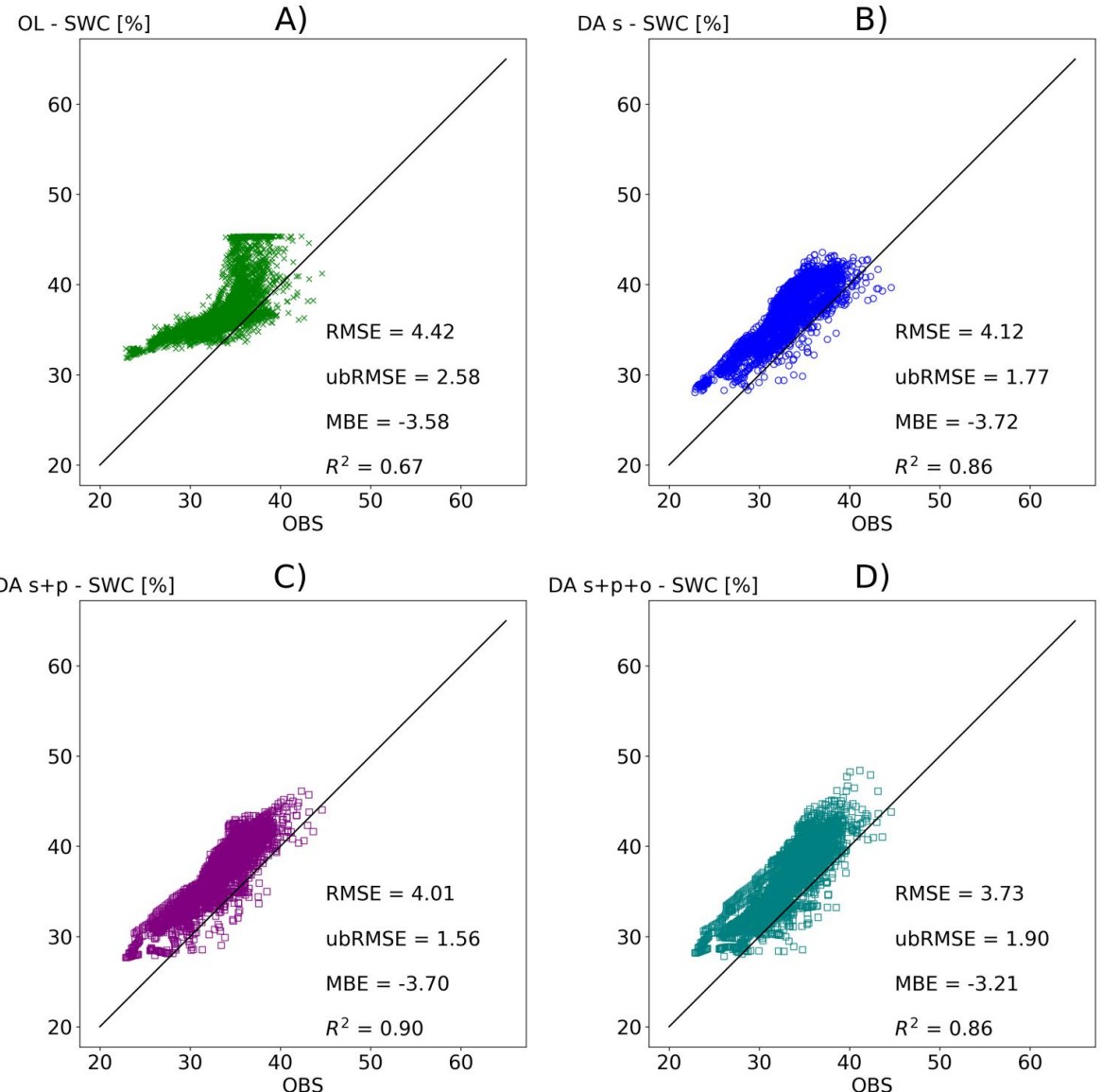

**Figure 6: Correlation diagrams for observed (OBS) and simulated soil water content (SWC) at 50 cm depth. Each marker shows one daily average. Subplot A) in the top left diagram shows open loop (OL), subplot B) in the top right shows data assimilation of state variable (DA_s), subplot C) in the bottom left shows data assimilation of state and parameters (DA_s+p), and subplot D) in the bottom right shows data assimilation of state and parameters including organic matter (DA_s+p+o). Each diagram shows root mean square error (RMSE), unbiased root mean square error (ubRMSE), mean bias error (MBE), and squared correlation coefficient (R²).**

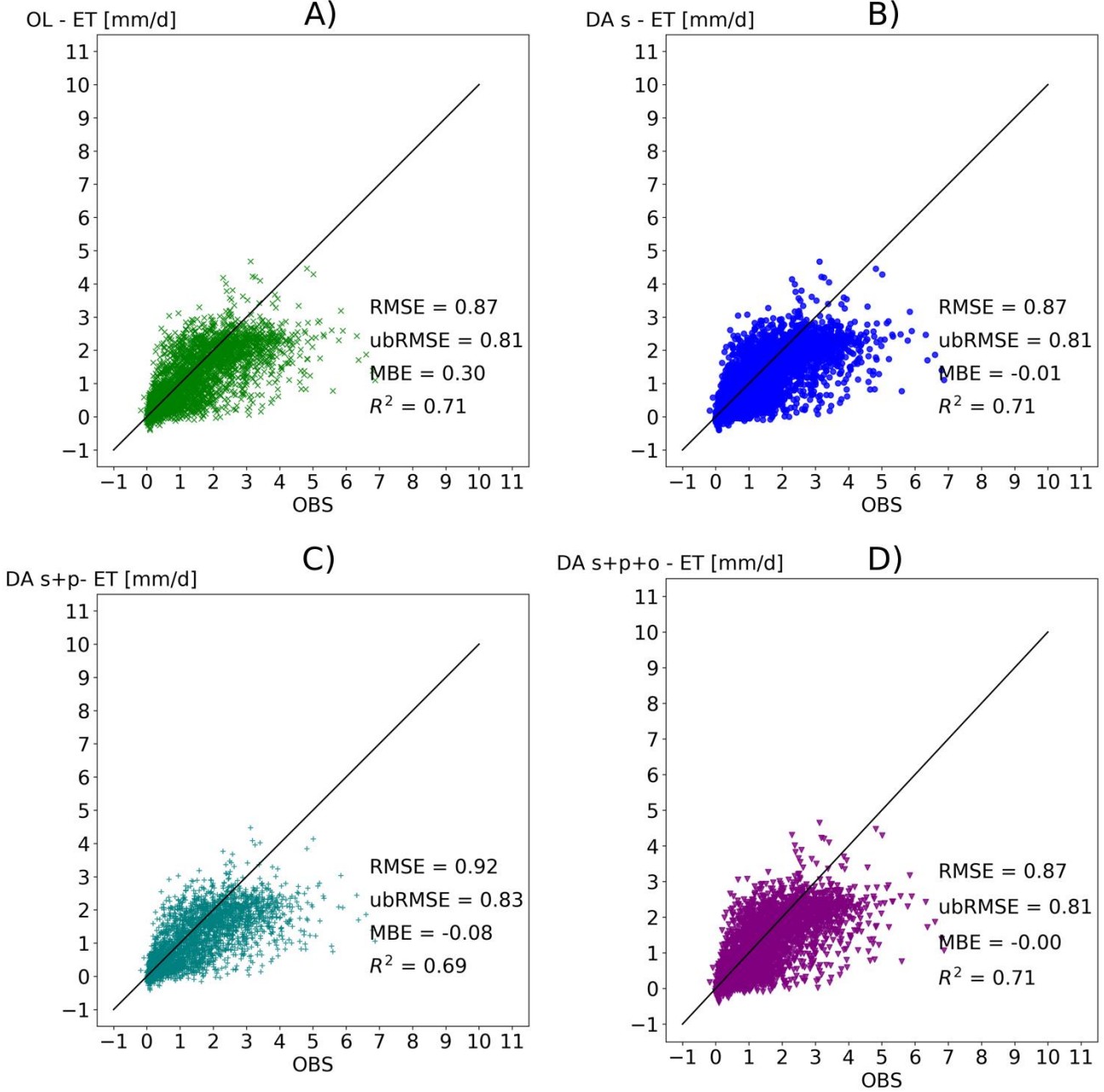

**Figure 7: Correlation diagrams for observed (OBS) and simulated evapotranspiration (ET) in millimeter per day [mm/d]. Each marker shows one daily average. Subplot A) in the top left diagram shows open loop (OL), subplot B) in the top right shows data assimilation of state variable (DA_s), subplot C) in the bottom left shows data assimilation of state and parameters (DA_s+p), and subplot D) in the bottom right shows data assimilation of state and parameters including organic matter (DA_s+p+o). Each diagram shows root mean square error (RMSE), unbiased root mean square error (ubRMSE), mean bias error (MBE), and squared correlation coefficient ($R^2$).**

**Table 1: Statistical properties and cross-correlation coefficients (CC) used to perturb the atmospheric forcing data.**

| | Perturbation | Mean | Standard deviation | CC **PR** | CC **SW** | CC **LW** | CC **TP** |
|---|---|---|---|---|---|---|---|
| Precipitation (**PR**) | Multiplicative log-normal distribution | 1.0 | 0.5 | 1.0 | -0.8 | 0.5 | 0.0 |
| Shortwave radiation (**SW**) | Multiplicative log-normal distribution | 1.0 | 0.3 | -0.8 | 1.0 | -0.5 | 0.4 |
| Longwave radiation (**LW**) | Additive normal distribution | 0.0 | 20.0 | 0.5 | -0.5 | 1.0 | 0.4 |
| 2m Air temperature (**TP**) | Additive normal distribution | 0.0 | 1.0 | 0.0 | 0.4 | 0.4 | 1.0 |

**Table 2: Statistical evaluation measures for the four different simulation and assimilation scenarios, always compared to measurements.**

| | OL | DA_s | DA_s+p | DA_s+p+o |
|---|---|---|---|---|
| RMSE / 5cm | 8.08 | 5.59 | 5.24 | 4.82 |
| ubRMSE / 5cm | 6.52 | 4.19 | 3.29 | 3.47 |
| MBE / 5cm | -4.78 | -3.7 | -4.07 | -3.35 |
| $R^2$ / 5cm | 0.63 | 0.91 | 0.93 | 0.92 |
| RMSE / 20cm | 4.03 | 2.84 | 2.27 | 1.67 |
| ubRMSE / 20cm | 3.56 | 1.62 | 1.86 | 1.64 |
| MBE / 20cm | 1.89 | 2.33 | 1.3 | 0.31 |
| $R^2$ / 20cm | 0.66 | 0.96 | 0.92 | 0.95 |
| RMSE / 50cm | 4.42 | 4.12 | 4.01 | 3.73 |
| ubRMSE / 50cm | 2.58 | 1.77 | 1.56 | 1.9 |
| MBE / 50cm | -3.58 | -3.72 | -3.7 | -3.21 |
| $R^2$ / 50cm | 0.67 | 0.86 | 0.9 | 0.86 |

**Table 3: Initial soil texture data and soil texture data after updating by data assimilation.**

| Type / depth | Initial ensemble mean | Updated ensemble mean | Updated ensemble standard deviation |
|---|---|---|---|
| Sand / 5 cm | 19.3 | 45.7 | 13.0 |
| Sand / 20 cm | 23.3 | 49.1 | 12.3 |
| Sand / 50 cm | 27.3 | 52.6 | 11.3 |
| Clay / 5 cm | 38.9 | 35.0 | 12.2 |
| Clay / 20 cm | 38.9 | 34.9 | 10.9 |
| Clay / 50cm | 37.9 | 33.4 | 10.5 |
| Organic matter / 5 cm | 34.1 | 51.4 | 8.17 |
| Organic matter / 20 cm | 15.8 | 32.3 | 7.8 |
| Organic matter / 50 cm | 8.7 | 13.1 | 4.9 |

670