# Peer review of "Coupling the Community Land Model version 5.0 to the parallel data assimilation framework PDAF: Description and applications"

_Geoscientific Model Development, 2021_

## Author Response (AR1)

**Scientific/Detailed Comments:** Referee comments in bold and author answer non-bold.

Review comments by Referee #1

**Line 16: Not clear to me how this coupling is 'novel'. The Data Assimilation Research Testbed (DART) uses similar ensemble capability, and couples an EnKF to CLM5. (Raczka et al., 2021 https://doi.org/10.1029/2020MS002421).**

Our intention of using the term 'novel' in this sentence was to highlight specifically the new coupling of CLM5 with PDAF in contrast to existing coupling of CLM5 to other data assimilation frameworks or previous versions of CLM to PDAF.

Changes in Text:
Lines: 17-18
"This newly implemented coupling integrates the PDAF functionality into CLM5 by modifying the CLM5 ensemble mode to keep changes to the pre-existing parallel communication infrastructure to a minimum."

**Line 28: Seems like a dated citation (Overgaard et al., 2006)– perhaps reference CMIP5 or CMIP6 manuscripts that compare a range of LSMs performance (e.g. Arora et al., 2020; https://bg.copernicus.org/articles/17/4173/2020/)**

We updated the dated citation with a more recent citation that highlights the complexity and range of current LSMs.

Changes in Text:
Lines: 30-31

"For example, see Arora et al. (2020) for a comparison of coupled atmosphere-land surface models in terms of projected carbon concentrations and carbon feedback  as part of the Coupled Model Intercomparison Project (CMIP)."

**Lines 34-36: Need to improve references to water limited regions and the work that has been done to improve water limitation and its connection to the carbon cycle. (e.g. Raczka et al., 2021; Weider et al., 2017; Kennedy et al., 2019) https://agupubs.onlinelibrary.wiley.com/doi/full/10.1002/2016JG003704).**

The text was modified.
Lines: 37-41

"Humphrey et al. (2021) shows that the inter-annual variability in land carbon uptake simulated by Earth system models is driven by anomalies in temperature and vapor pressure deficit, which are controlled by soil moisture variability. However, they conclude that the partitioning between direct and indirect soil moisture effects is more dependent on modeling approaches and that more physical and holistic modeling of the vegetation response to drought could reduce uncertainties in climate projections.

Lines: 43-45
"The latest version, CLM5, is especially of interest because it includes various improvements over previous versions. For example, Kennedy et al. (2019) implemented a new plant hydraulic stress

parameterization and showed improvements in simulating transpiration and soil water content of a tropical forest site.“

Lines: 55-57
“Wieder et al. (2017) used CLM4.5 to investigate the impact of extending growing seasons on carbon, water, and energy fluxes and found that of the five ecosystems considered, wetland ecosystems were the most affected.”

Lines: 91-92
“Raczka et al. (2021) used DART to assimilate remotely sensed leaf area index and above ground biomass in CLM5 to improve carbon flux simulation”

**Line 35-40: In general, citations provided here seem rather general, and not focused on particular research topic, which in this case is SWC, hydrology and impact upon latent and sensible heat. You might want to focus more directly on the representation of hydrology within CLM (Swenson et al., 2019; https://agupubs.onlinelibrary.wiley.com/doi/full/10.1029/2019MS001833 ), and why it suits DA for your application. Including specific advances in hydrology with CLM5.0 (Kennedy et al., 2019; https://agupubs.onlinelibrary.wiley.com/doi/10.1029/2018MS001500) might also better motivate this work.**

It is true that the references here are rather general. We chose them to highlight the application of CLM to single-point setups for a wide array of studies.

Changes in Text:
Lines: 45-48

“In addition, Swenson et al. (2019) improved CLM5 further by implementing lateral flow, i.e., water fluxes within a CLM5 grid cell between soil columns with different slopes, reproducing differences in evapotranspiration between upland and lowland hillslopes. In many studies, comparisons of CLM model results were made with in situ observations using a single grid cell setup.”

**Line 45: Need more background into remote sensing products of soil moisture. There are a growing set of remotely-sensed soil moisture observations that should be referenced here – SMOS, SMAP, ESA-CCI. There are many emerging products. You should better motivate the use of DA precisely because the range of remotely-sensed products is expanding. Also the purpose of DA (especially EnKF) is that unobserved states (subsurface layers) can be adjusted based upon the model state covariance matrix of the modeling system.**

The emerging remotely-sensed soil moisture observations products are important and useful in combination with DA. We did not include many references to remotely-sensed products because our application used only in-situ measurements. However, we agree that remotely-sensed products are a good way to motivate DA applications and we added a reference to the new satellite products.

Changes in Text:
Lines: 61-64

“Nevertheless, a growing number of soil moisture products from remote sensing has become available e.g. Soil Moisture and Ocean Salinity (SMOS) (Kerr et al., 2010), Soil Moisture Active

Passive (SMAP) (Entekhabi et al., 2010), European Space Agency Climate Change Initiative (ESA-CCI) (Dorigo et al., 2017), which are used to improve the accuracy of land surface model predictions, e.g. of soil moisture, energy and carbon fluxes, through data assimilation."

**Line 47: Awkward sentence, It is common practice....**

We rephrased the sentence.

Changes in Text:
Lines: 67-68

"It is common practice that numerical models are implemented without intrinsic data assimilation and external frameworks are used to perform data assimilation."

**Line 53: It is unclear what distinction is being made between 'offline' vs 'online' coupling in data assimilation frameworks. The authors state that 'offline-coupled' data assimilation is used for the Data Assimilation Research Testbed (DART) https://dart.ucar.edu/ (Anderson et al., 2009). Furthermore, in offline coupling 'the framework wraps around the model and does not modify the model'. This is not true. Within DART – the state of the model is modified during the update step of the EnKF. Therefore, the assimilation updates the model state in time so that the trajectory of the model more closely matches the observations being assimilated. Furthermore, DART and CLM are interactive in that DART updates the model state within CLM, and the inflation parameters within DART (Gharamti et al., 2019; doi.org/10.1175/MWR-D-20-0101.1) are also updated in time and influence the ensemble spread of the CLM model state. More recent updates to the CLM code have included model components that call data assimilation components from DART directly. In applications outside of CLM, DART has been used to modify parameters (e.g. Zhang et al., 2021; doi.org/10.5194/tc-15-1277-2021.) within models as well. The authors need to reconsider their assertion that their DA coupling approach is 'novel'.**

We acknowledge that the distinction between 'offline' vs 'online' is not clear. We use the definition of these terms in this context from PDAF as formulated for example in Kurtz et al. 2016. The distinction is about using the main memory or restart files for the transfer from model states to the DA framework and vice versa. We corrected the 'the framework wraps around the model...' sentence. We did not mean to imply that offline coupling does not affect model states or that DART and CLM are not interactive but rather that in offline coupling it is often not necessary to modify the model source code. We were not aware of the recent updates to CLM that includes calls to DART directly since even recent publications like Raczka et al., 2021 and Zhang et al., 2018 still mention the use of restart files for the exchange between the model and DART. We corrected the comparison to DART in this section to include your corrections. We did not intend to assert that the DA coupling approach is novel, we specifically mention that we mostly re-use and modify existing software infrastructure, we just meant that our specific implementation of coupling CLM5 and PDAF is 'novel'.

Changes in Text:
Lines: 72-74

"We can distinguish between two different approaches for the coupling of models with external frameworks. In case of offline coupling, the framework wraps around the model and does not modify the model source code but instead interfaces with the model through output files."

Lines: 80-81
"While the studies cited in this section use DART for offline coupled data assimilation, we were made aware that the use of DART for online coupling is in development."

Lines: 101-102
"In general, online coupling is important in high performance computing to avoid time consuming file read/write operations."

**Line 57: DART is commonly used with all components of the earth system within CESM including land (CLM), atmosphere (CAM), ocean (POP), and sea/land ice, as well as many other earth system models. See https://dart.ucar.edu/publications/**

We will modify the sentence and mention the use of DART in other CESM components and other earth system models.

Changes in Text:
Lines: 77-80

"The Data Assimilation Research Testbed (DART) (Anderson et al., 2009), which was originally developed for data assimilation with atmospheric models, is commonly used for offline coupled data assimilation with all components of the Earth system within the Community Earth System Model, including land, atmosphere, ocean, sea/land ice, and other earth system models."

**Line 84: "... [these manuscripts] concluded that the consideration of heterogeneous porosities can increase model performance depending on the model structure. In contrast to these detailed distributed catchment studies, we model the study site from the viewpoint of a larger regional model where the catchment is represented by a single grid cell."**

**The previous modeling studies suggest that including a description of heterogeneous soil porosities will help model performance in a fine-scale catchment. Presumably a fine spatial scale description is needed to represent a catchment. Therefore it caught this reviewer off-guard that the authors propose to use a coarse, grid cell to represent catchment behavior(see Swenson et al., 2019). Perhaps provide motivation that a DA framework can improve modelling behavior through correcting for known biases in the system – or known errors in parameters.**

Yes, representing a catchment in detail requires a fine spatial scale simulation, however in many applications it is not computationally feasible to represent catchments on such fine scales. Therefore, we think it is reasonable to demonstrate the application of a new DA coupling on the coarse scale of many applications but simplified to a single grid cell to highlight the direct effects of DA for the grid cell where observations are available. We agree that we can better motivate the DA framework and we rephrased this section.

Changes in Text:
Lines: 116-121

"However, in earth system modeling applications, distributed simulation of such small catchments is usually computationally not feasible and a single grid cell is used instead. With such coarse scale applications in mind, and to demonstrate the application of CLM5-PDAF in a simplified model setup, we represent the Wüstebach catchment by a single grid cell. Furthermore, using a single grid cell approach, we can showcase the improvements data assimilation and parameter updating can provide for correcting biases in the system and errors in the parameters."

**Method section 2.1**
**"Furthermore, we investigate whether updating of the soil organic matter parameter via data assimilation can further improve the prediction of soil water with CLM5."**
**Given your manuscript goals you need to provide some explanation within the CLM methods section of how soil organic matter influences soil water drainage. It is also slightly unclear what benefits either updating to the CLM 5 description or using the PDAF will bring to this analysis, be a bit more specific. Some of this information is included in the appendix (A1-A4), but a bit more explanation within the main text would be helpful. May also want to mention in the methods of CLM 5 – that it updates the plant hydraulic stress representation (Kennedy et al., 2019) thereby influencing water-carbon coupling, and transpiration. The authors do not really discuss the influence of vegetation (water- carbon coupling) upon their SWC results.**

We added more details about the CLM5 soil organic matter parameter and its relation to SWC into the main text and included the relations to the plant hydraulic stress representation in CLM5.

Changes in Text:
Lines: 133-134

"The new plant hydraulic stress parameterization by Kennedy et al. (2019) impacts both the soil water content and also the coupling to the carbon cycle."

Lines: 144-152
"Starting with the version 4 of CLM the hydraulic parameters are also depending on organic matter content in the soil. For example, without the contribution of organic matter, the soil porosity in CLM is limited to a maximum of 0.489 for soils without sand fraction due to the implemented pedotransfer function of Clapp and Hornberger (1978). However, as can be seen in Figure 3, the soil water content observations in the Wüstebach catchment show frequently higher values. Incorporating the new equations with soil organic matter content increased the maximum value for porosity at the surface to 0.93 with decreasing porosity values with increasing soil depth. This shows that in order to simulate soil moisture in forest soils with high porosity, it is important to consider organic matter. The detailed equations for accounting for organic matter on soil hydraulic parameters can be found in Appendix A."

**Section 2.2.1**
**"For example, ensemble members can be generated based on perturbed soil parameters and atmospheric forcings." Not clear at this point how ensemble is generated for this experiment. "The state vector contains soil water content (model states), sand and clay fractions (parameters), and organic matter fractions (parameters) depending on the experiment as described in Section 3.3." It makes sense here to describe how the CLM soil column is constructed (i.e. PFTs, columns, layers etc) within Section 2.1. Are you updating all soil layers of the CLM model for SWC?**

We added a reference to section 3.2.2 and 3.2.3 for the specifics of ensemble generation for this specific experiment. We added details of the CLM soil column structure as you suggested. It is

correct that we are updating the SWC of all soil layers of the CLM model and added this in the revised version.

Changes in Text:
Lines: 187-188

"The perturbations of soil properties and forcings represent the uncertainty range of the model, the specifics of the ensemble generation for this study are described in Sections 3.2.2 and 3.2.3"

Lines: 194-198
"CLM5 uses a subgrid hierarchy that contains land units, columns, and patches. Patches represent different plant functional types and share a single column. The physical state variables, like soil water content, are defined at column level and vertically discretized into layers. There are up to 20 hydrologically active layers depending on the depth to bedrock parameter. For simplicity, we consider the model state for soil water content to be the 20 layers of the column even if not all 20 layers are active."

**Section 2.3:**
**"Furthermore, for the optional parameter updating it is necessary to provide a function to transform the input parameters, e.g. soil texture, to the model parameters, e.g. the soil hydraulic parameters. CLM5 performs this transformation once during initialization to obtain the hydraulic parameters from the soil texture in the surface file."**
**It was a bit confusing to this reviewer that the authors were referring to the soil characteristics such as clay/sand/organic matter as 'parameters'. In general, parameters refer to numeric coefficients that influence model equations. This manuscript adjusts the soil characteristics to indirectly adjust the hydraulic parameters (A1-A4). In general, it seems parameter optimization should be limited to parameters which are difficult/impossible to measure. The soil characteristics, on the other hand, could be measured given how well the study site (watershed) seems to be observed already.**

We refer to the soil characteristics as parameters because we treat them as parameters in the joint state and parameter approach. As we describe in the outlook, we agree that the parameter optimization should be applied to the hydraulic parameters directly. Using the soil characteristics as indirect parameters to be updated has been done in various previous studies and was therefore the baseline implementation. Nevertheless, we mention in the revised version that the soil hydraulic parameters will be updated directly in future works.
In addition, we disagree that parameter estimation should be limited to parameters difficult to measure. Hydraulic conductivity can be measured, but shows even at the small scale often a large spatial variability and is therefore still an important source of uncertainty for model simulations.

Changes in Text:
Lines: 209-211

"A more consistent approach would be to update the hydraulic parameters directly instead of updating the soil characteristics and using the pedotransfer function. However since the existing implementations uses the indirect approach we chose to follow the same approach in this study."

**3.1 Study Site: Very unclear how the CLM site-level or gridded simulation was setup. What was the size of the grid cell used in which the soil characteristics / topography were defined? How was this forested site initialized? Was it spun-up from near ground conditions or was a present-day compset used within CLM?**

The grid cell size used was roughly 3km by 3km. The model was initialized from a cold start and spun up according to the CLM5 documentation. We added more details about this.

Changes in Text:
Lines: 300-304

"For the modeling, we use a grid cell size of 3 by 3 km based on the grid used in Naz et al. (2019) for a continental scale study. Unless specified, we used the default parameters of CLM5 and followed the instructions of the online CLM5 user guide to get initial soil characteristics, topography, and other initial parameters of the surface file. The model was spun-up from a cold start as described in the CLM5 user guide with atmospheric forcings from 2009 to 2018 described in more detail in Section 3.2.2. More specific details on the different simulation setups are presented in Section 3.3. "

**Section : 3.2.1**
**"The filtered raw data is then spatially and temporally averaged to fit the requirements of the model, i.e., daily averages for the three soil depths." I don't think that's a limitation or requirement of the model – CLM5.0 can be run on an hourly time basis thus assimilation could be performed hourly. Also there are roughly 25 subsurface potential soil layers in CLM, so it could potentially handle more soil depth observations depending upon the depth of the soil column at this location. I think you performed daily averages of all the soil observation locations to simplify the assimilation process, which is reasonable. So you averaged all the forested (undisturbed) soil water observation locations into a single value for each depth?**

You are right, it is not a requirement or limitation of the model. We wanted to say that these are the requirements we defined for this model setup. We corrected this sentence accordingly. Yes, we averaged the large number of forested soil water observation locations into a single value for each depth (at 5, 20 and 50 cm, the three depths for which observations were available) and specified this in the revision.

Changes in Text:
Lines: 315-317

"The filtered raw data is then spatially and temporally averaged to fit our setup of the model, i.e., daily averages for the three soil depths from the average of the selected stations."

**Line 261: Lateral flows are not represented at all in CLM5 – no grid cell to grid cell communication. Surface and subsurface drainage is routed directly to rivers.**

Yes, we just meant that lateral flow in the form of runoff to rivers is represented in CLM5.

Changes in Text:

Lines: 320-324

"However, lateral flows are only represented through routing to rivers in CLM5. Therefore, we omitted the riparian zone and selected only SoilNet stations located in the groundwater distant forested parts of the Wüstebach catchment in this study."

**Line 287: Be more specific here: Perturbed inputs of \*both\* atmospheric forcing and soil characteristics of soil/clay and organic matter? What was the purpose of perturbing both? Could you use only atmospheric perturbations if the goal was to only assimilate SWC observations? The additional perturbation of the soil/clay, organic matter was necessary for the parameter updates? Provide a bit more explanation.**

Yes, we used perturbed inputs of both atmospheric forcings and soil characteristics. We perturb both because the perturbation determines the ensemble spread and the ensemble spread represents the model uncertainty. Model uncertainty exists for both atmospheric forcings and the model parameters i.e. the soil hydraulic parameters indirectly determined through the soil characteristics. Even in studies without parameter updates and only SWC observations soil characteristics are often perturbed. We added these explanations in the revision.

Changes in Text:
Lines: 345-347

"These simulations are equivalent to CLM5 standalone ensemble simulations with perturbed inputs for both atmospheric forcings and soil characteristics. The perturbed inputs represent both forcing and model uncertainty and determine the ensemble variance."

**Line 287: Do you state anywhere what soil water variable in CLM you are adjusting? I assume it is the prognostic variable H2OSOI_LIQ, but there is also H2OSOI_ICE and the diagnostic variable H2OSOI. Also you are adjusting all vertical layers?**

We did not specifically mention the CLM5 variable names. We used the diagnostic variable H2OSOI as a state variable and then adjust both prognostic H2OSOI_LIQ and H2OSOI_ICE variables for all vertical layers. We made this clearer in the revision.

Changes in Text:
Lines: 198-200

"Specifically, we use the diagnostic soil water content variable called "H2OSOI" as the model state variable and after each update adjust the prognostic liquid and solid water content variables "H2OSOI_LIQ" and "H2OSOI_ICE"."

**Figure 1: Any physical explanation of why the model would overestimate SWC at shallow depth (5 cm) and at the deepest layer (50 cm), but overestimate SWC at the middle depth (20 cm)? Curious of whether this could be related to the observational uncertainty of the SWC sensor – and what was used as the observation uncertainty? Also wondering if this behavior was related to the configuration of the root profile within CLM – how much of the root mass was within this layer and therefore what influence this had upon transpiration and removal of water within this soil layer?**

We currently do not have a physical explanation of this model behavior. For simplicity, the observational uncertainty is assumed be constant and set to a RMS of 2%. We do not expect that the behavior is related to observational uncertainty, as measurements from a large number of measurement locations were averaged. We did not explore the effect of the root profile on this behavior. We clarified this in the revision.

Changes in Text:
Line: 365-367

"The simulations tend to have a wet SWC bias compared to the observations at 5 and 50 cm depths but underestimate SWC at 20 cm depth. This behavior could be the result of the root profile in CLM5 or other uncertainties related to model parameters."

**This opens up other questions of what the forest state was for your model simulations including things like biomass and leaf area index from the site observations. Were these reasonable? Did you look at the simulated transpiration, evapotranspiration and GPP to determine if these values seemed reasonable? I don't think you had flux tower observations available to check, but perhaps you could infer reasonable values from surrounding sites. The vegetation state will have an important impact of subsurface soil moisture state and to what effect this impacted your simulation is unclear. The vegetation state, including how it was initialized and how it was simulated (other than the PFT setting) was not discussed in this manuscript.**

We looked at LAI and evapotranspiration and they are reasonable. For ET even close to flux tower observations. We have now included a figure for ET that shows that SWC DA did not have a significant impact on ET simulation and more analysis is required. We are currently in the process of performing more simulations for different sites and analyzing the effects of SWC DA on ET and other variables in a further study. For this study, we mainly wanted to demonstrate the coupling of PDAF to CLM5 and the direct and clear impact of SWC DA for a simple setup. We added a short discussion to the revision.

Changes in Text:
Lines: 389-392

"Figure 7 shows the impact of the soil water content data assimilation on the evapotranspiration flux (ET). For all statistical characteristics, the impact is negligible for the three data assimilation scenarios. We believe this is due to the overall wetness of the study area, i.e.,  the soil water content is not the limiting factor for, so other variables or parameters would need to be assimilated to affect simulated evapotranspiration."

Lines: 417-419
"We were not able to show significant impact of the assimilated soil water content on evapotranspiration flux.  In a future study, we will investigate whether this behavior also occurs for study sites in other climates. We will also include other variables and parameters in the data assimilation to test their effects on the evapotranspiration flux."

**Table 3: It was not completely clear until I viewed this table that the model 'parameters' that were being adjusted within the assimilation were actually the soil characteristics of clay/sand and organic matter. The term 'parameter' is admittedly loosely defined in modeling applications, but in general, this typically refers to 'coefficient' values within the model code**

**that (within a model like CLM) are specific for particular plant functional types. The surface characteristics of the soil, however, are typically prescribed and held constant. The reviewer recognizes that this manuscript is, in part, is a demonstration of the capabilities of the assimilation system, and is apparently following the approach taken in (Naz et al., 2019) but physically, does it make sense to adjust the soil characteristics (generally fixed in time) such that they change with time? Would it not make more sense to adjust the numeric coefficients in equations A1-A4 instead of %sand and %clay? The authors acknowledge this at the very end of the conclusion section, but perhaps more justification could be provided earlier on in the manuscript.**

**If there were many soil moisture subsurface observations, were any soil characteristic observations available to check the posterior values of the soil characteristics?**

As mentioned in the previous question, we will be more careful with the terminology "soil characteristics" and "soil parameters". We used the term 'parameter' more generally as a distinction to state variables rather than to differentiate between coefficients and prescribed constants. We fully agree that it makes more sense to adjust the coefficients in A1-A4 directly. The indirect approach using the soil characteristics was an established approach, but in future work we will adjust the numerical coefficients directly. As you suggest, we mention this earlier in the manuscript and not just in the conclusions. There are soil characteristic observations at different locations in the catchment, but it is not a simple task to 'average' these discrete spatially distributed observations to compare it to the posterior soil characteristic value that represents the whole catchment / grid cell. We added the discussion on this in the revision.

Changes in Text:
Lines: 209-211

"A more consistent approach would be to update the hydraulic parameters directly instead of updating the soil characteristics and using the pedotransfer function. However since the existing implementations uses the indirect approach we chose to follow the same approach in this study."

Lines: 396-400
"For this particular study site, there exist measurements for the soil characteristics at various points throughout the catchment. However, we did not perform comparisons between the updated soil characteristics values and measurements, since it is not simple to overcome the heterogeneity of discrete spatially distributes point measurements of soil characteristics and scientifically combine them into a coarse catchment scale value."

**Major Comments**
**The motivation for this study is weak. The authors briefly mention about the difference between online and offline DA (Ln 55), but they need to better motivate the coupling CLM5.0 with PDAF. Is it more for the standalone DA with CLM5.0 or for CLM5.0 within the TSMP framework? What new does PDAF bring? How does it reduce the number of core-hours or computation time compared to other offline DA? And, how it scales with increase in domain size and time period of simulation? This needs to be discussed clearly.**

It is both for the standalone DA with CLM5, as shown in this study, and also the potential for future use in the complete TSMP framework. We intended to motivate this in the introduction, which has been modified. We did not perform computational performance comparison to other DA frameworks for this specific application. General scaling behavior of PDAF has been presented in Nerger et al. (2013) and Kurtz et al. (2016). Without extensive computational studies we do not want to discuss advantages and disadvantages of different DA frameworks in detail. Instead we want to focus on the specific implementation and application of a new coupling that can be used to perform DA with CLM5. We mention this aspect in the revised version.

Changes in Text:
Lines: 99-103

"In this study, we present the coupling of PDAF as a framework for the data assimilation because it provides many data assimilation algorithms, supports online coupling, and includes templates for the modifications to the model code that are necessary for the coupling with CLM5. In general, online coupling is important in high performance computing to avoid time consuming file read/write operations. In this regard, Nerger et al. (2013) and Kurtz et al. (2016) have demonstrated the excellent scaling and performance of PDAF, for which reason we selected PDAF for our data assimilation study with CLM5."

**Kurtz et al. (2016) already presented the PDAF coupling to TSMP including CLM3.5. So, what is new in this study? I assume that there must be substantial work involved in developing the PDAF interface around CLM5.0 which has different software environment compared to earlier versions of CLM (e.g. CLM3.5). But it is not so clear in the current version of the manuscript.**

The new developments in this study are the modifications to what Kurtz et al. (2016) presented. These modifications are necessary to interface with CLM5.0 which, as you mentioned, has a different software environment compared to earlier versions. We discuss the implementation and differences in section 2.3 in detail, we made the differences clearer and highlight the new developments more in the revised version.

Changes in Text:
Lines: 106-108

"The new developments in this study for integrating CLM5 into the TSMP environment include changes to the interface to CLM5 and a new software environment, which are described in detail in Section 2.3."

Lines: 231-233

"Figure 1 sketches the organisation of the CLM5-PDAF coupling into five main components. The next paragraphs describe these components in more detail and modifications compared to the CLM3.5-PDAF implementation by Kurtz et al. (2016) are discussed."

Lines: 234-235
"The new code developments in the PDAF user functions are superficial inclusions of CLM5 as option with the same functionality as already implemented and described by Kurtz et al. (2016) for CLM 3.5."

Lines: 240-242

"Therefore, the TSMP wrapper contains the modified routines from the model for initialization, time stepping, and clean-up. The development of CLM5-PDAF includes modifying these routines from the original CLM5 source code."

Lines: 252-253

"Additionally, the TSMP wrapper contains the model specific routines for managing the PDAF state vector. As these routines are model dependent, part of the development of CLM5-PDAF included the creation of routines to interface with CLM5."

**Ln 85: This comes so suddenly. The authors need to provide a better motivation to use a single column model. The literature review is another weak part of the manuscript. The authors make no effort in presenting their results in context of previous findings. Also, does the improvement in soil moisture also improves the surface energy fluxes. For LSMs, improvements need to be explored soil states as well as fluxes. And, a discussion section is missing.**

We provide context for this study in the literature review for single-point studies (lines 48-57), for data assimilation in LSMs (lines 81-92), for the specific software framework (line 94-98), and for the specific site (line 110-116). We added more literature references and make the respective contexts clearer in the revised version.

We included results and discussion for the DA-related ET changes in the revised version. We added a discussion section.

Changes in Text:
Lines: 47-48

"In many studies, comparisons of CLM model results were made with in situ observations using a single grid cell setup."

Lines: 55-57
"Wieder et al. (2017) used CLM4.5 to investigate the impact of extending growing seasons on carbon, water, and energy fluxes and found that of the five ecosystems considered, wetland ecosystems were the most affected."

Lines: 116-121
"However, in earth system modeling applications, distributed simulation of such small catchments is usually computationally not feasible and a single grid cell is used instead. With such coarse scale applications in mind, and to demonstrate the application of CLM5-PDAF in a simplified model setup, we represent the Wüstebach catchment by a single grid cell. Furthermore, using a single grid

cell approach, we can showcase the improvements data assimilation and parameter updating can provide for correcting biases in the system and errors in the parameters."

**There is no README file or User manual to reproduce the results presented in this study, also please provide a web URL for Zenodo and cite this paper in the References. The upload should also include scripts for processing the figures and observation data for reproducibility.**

We created a README, and added scripts for processing and observation data in the revised version.

Changes in Text:
Lines: 451-452

"Code availability.
The development branch of the CLM5-PDAF coupling is freely available via Zenodo, 10.5281/zenodo.5720866 or https://zenodo.org/record/5720866 ."

Lines: 593-595
"Strebel, L., Bogena, H., Vereecken, H., and Hendricks Franssen, H.-J.: README for the source code of "Coupling the Community Land Model version 5.0 to the parallel data assimilation framework PDAF: Description and applications" https://zenodo.org/record/5720866 (2021)."

**Minor Comments**

**Ln 10: Even tuned second generation LSMs can be "accurate", here maybe the authors want to imply that third generation LSMs better represent the key physical processes. Also, check in the rest of the manuscript.**

Yes, we intended to stress the improvements in the representation of physical processes. We modified the sentence to make this clear.

Changes in Text:
Lines: 9-11

"They are continuously improving and becoming  better in representing the different land surface processes, e.g. the Community Land Model version 5 (CLM5)."

**Ln 11: more? What type of data?**

Various types, from new satellite products to new in-situ measurement stations, also new cosmic-ray and flux tower sites. We added this information in the revised version.

Changes in Text:
Lines: 11-13

"Similarly, observational networks and remote sensing operations are increasingly providing more data, e.g. from new satellite products and new in-situ measurement sites, and also higher quality data for a range of important variables of the Earth system."

**Ln 15: Is this further development of PDAF or addition of new interface to connect PDAF with new models?**

It is the addition of a new interface to connect PDAF with a new model, we corrected this sentence in the revised version.

Changes in Text:
Lines: 16-18

"In this study, we present the development of the new interface between PDAF and CLM5. This newly implemented coupling integrates the PDAF functionality into CLM5 by modifying the CLM5 ensemble mode to keep changes to the pre-existing parallel communication infrastructure to a minimum."

**Ln 34: common might not be the right word here.**

We corrected the sentence.

Changes in Text:
Lines: 41-42

"A commonly used LSM is the Community Land Model (CLM) (Lawrence et al. 2019), of which the performance has already been evaluated in various studies with observational data."

**Ln 48-53: This paragraph needs to be rephrased (framework, external framework, within framework). It has just too many frameworks.**

We rephrased the paragraph with fewer 'frameworks'.

Changes in Text:
Lines: 69-72

"Coupling to such external codes instead of implementing data assimilation inside the numerical model provides many advantages. External frameworks are usually built for modularity and extendibility, i.e., they provide multiple different data assimilation methods and can be updated when new methods are developed. Additionally, external frameworks are usually optimized for parallel computing."

**Ln 70: PDAF with joint state parameter update for CLM was also used in the following study: Shrestha, P., W. Kurtz, G. Vogel, J.-P. Schulz, M. Sulis, H.-J. Hendricks Franssen, S. Kollet and C. Simmer (2018), Connection Between Root Zone Soil Moisture and Surface Energy Flux Partitioning Using Modeling, Observations, and Data Assimilation for a Temperate Grassland Site in Germany. JGR-Biogeosciences doi: 10.1029/2016JG003753**

We included this reference as another example of joint state parameter update with PDAF and clm3.5 in the revised version.

Changes in Text:
Lines: 94-96

"For example, Shrestha et al. (2018) successfully used PDAF to perform joint state and parameter updates with CLM3.5 to improve soil moisture prediction and suggested that this approach is applicable to CLM5."

**Ln 73: "In this study, we present the coupling of .."**

We rephrased the sentence.

Changes in Text:
Lines: 99-101

"In this study, we present the coupling of PDAF as a framework for data assimilation because it provides many data assimilation algorithms, supports online coupling, and includes templates for the modifications to the model code that are necessary for the coupling with CLM5."

**Ln 93: Rephrase. "The paper ends with " is not appropriate.**

We corrected the sentence.

Changes in Text:
Lines: 127-128

"We end with a discussion, conclusions, and an outlook on further planned improvements, for example concerning parameter updating."

**Ln 116: 1) variation methods, ...2) sequential methods**

The reference we cite (Reichle 2008) calls them 'variational methods'. We added the numbering.

Changes in Text:
Lines: 158-159

"In Earth sciences, two common data assimilation approaches are 1) variational methods, often used in atmospheric models, and 2) sequential methods like the Ensemble Kalman filter (Reichle 2008)."

**Ln 125: Perturbation vector missing in Eq. 1, where y is generally the observation vector. It is discussed much later in Ln 146. What is the measurement error?**

We moved the inclusion of the perturbation to the observation vector closer to Eq. 1. For simplicity, the measurement error is assumed to be constant and set to a RMS of 2%. We mention this in the revised version.

Changes in Text:
Lines: 172-174

"where the superscript i refers to ensemble member i, $x_a^i$ is the updated state vector after the analysis, $x_f^i$ is the forecasted model state vector, **K** is the Kalman gain, **y** is the observation vector, and **H** is the so-called measurement operator that transforms between model and observational states. Observational data is perturbed for each ensemble member to maintain the correct error statistics (Burgers et al. 1998). Therefore, y in equation 1 is shorthand for y=o+i, where o is the observational data and i is a perturbation vector with mean zero and covariance according to the observational error covariance matrix. For simplicity, the observational error is assumed to be constant and set to a root mean square of 2%."

**Section 2.3: There is always a discussion about older version, maybe the authors should discuss it before, and present their new formulation, rather than interchanging now and then. Maybe this would also highlight, what new work has been done.**

We compare to the coupling with the older version of CLM three times in this section: 1) In the section about the difference in time stepping between CLM5 and TSMP. Here the comparison is not strictly necessary, but highlights that the approach to modify the driver is the same as before even if the software environment has changed significantly. 2.) To point to the changes in CLM5 hydraulic parameter calculations, which includes the new changes with the addition of soil organic matter. 3.) To mention that the more complex software environment motivates the modification of the existing CLM5 ensemble mode.
Other comparisons in the section are not to older versions but to the framework of TSMP or PDAF specifically. We think it would be less useful to separate any comparisons, since they are mostlyused to give context to new implementations. Nevertheless, we highlighted more clearly the new work that has been done in the revised version.

Changes in Text:
Lines: 231-233

"Figure 1 sketches the organization of the CLM5-PDAF coupling into five main components. The next paragraphs describe these components in more detail and modifications compared to the CLM3.5-PDAF implementation by Kurtz et al. (2016) are discussed."

Lines: 234-235
"The new code developments in the PDAF user functions are superficial inclusions of CLM5 as option with the same functionality as already implemented and described by Kurtz et al. (2016) for CLM 3.5."

Lines: 240-242

"Therefore, the TSMP wrapper contains the modified routines from the model for initialization, time stepping, and clean-up. The development of CLM5-PDAF includes modifying these routines from the original CLM5 source code."

Lines: 252-253

"Additionally, the TSMP wrapper contains the model specific routines for managing the PDAF state vector. As these routines are model dependent, part of the development of CLM5-PDAF included the creation of routines to interface with CLM5."

Lines: 241-242
"The development of CLM5-PDAF includes modifying these routines from the original CLM5 source code."

**Ln 181: The "Figure 1" is not helpful, either improve or remove. Also, rephrase and elaborate the discussion.**

We use Figure 1 as a visual aid to describe the structure of both the actual implementation and the paragraphs in the section. We improved the figure by adding more information and elaborated the discussion.

Changes in Text:
Line: 231-233

"Figure 1 sketches the organization of the CLM5-PDAF coupling into five main components. The next paragraphs describe these components in more detail and modifications compared to the CLM3.5-PDAF implementation by Kurtz et al. (2016) are discussed."

[Figure]

Figure 1: Components of TSMP CLM5+PDAF highlighting the distinct separation of PDAF functionality, TSMP driver and wrapper, and CLM5 pseudo-library. Modifications are in relation to the implementation by Kurtz et al. (2016).

**Ln 191: What is "CIME"?**

CIME is the default clm5.0 driver. We added a definition in the revised version.

Changes in Text:
Lines: 242-243

"These routines are moved from the CLM5 default  driver, which is taken from the Common Infrastructure for Modeling the Earth (CIME) framework, into the TSMP wrapper. "

**Ln 204: Maybe "clipping" ?**

We corrected the sentence in the revised version.

Changes in Text:
Lines: 258-259

"The subroutine to update the state vector contains functionality to detect and correct invalid values, e.g. below residual soil water content, above porosity, and below 0% or above 100% for the sum of the sand and clay fractions."

**Ln 218: Rephrase.**

We rephrased the sentences.

Changes in Text:
Line: 272

"In ensemble simulations each member has individual input files."

**Ln 232: in Wüstebach, and Belgium ?**

We corrected the sentence.

Changes in Text:
Lines: 286-287

"The coupled modeling framework is applied to the small (38.5 ha) forested catchment called Wüstebach which is located in the Eifel National Park near the Belgium-Germany border."

**Ln 252: Explain the SWC unit.**

We added a definition for the volumetric soil water content.

Changes in Text:
Line: 308-313

"The observational data is the soil water content, i.e. the ratio of the volume of water to the porosity. This data is pre-processed using filters that remove data points based on their quality flag, spikes, frozen soil condition, and erroneous values. Spikes are defined as reductions in soil water content of more than 1% or increases in soil water content of more than 5% with an immediate return to values within 1% of the value before the spike. Soil water content below 1% or above 90% is considered erroneous."

**Ln 305: "to overestimate SWC" or "wet bias in SWC"**

We rephrased the sentence.

Changes in Text:
Lines: 365-366

"The simulations tend to have a wet SWC bias compared to the observations at 5 and 50 cm depths but underestimate SWC at 20 cm depth."

**Ln 333: What is variant here?**

The online variant to differentiate from the offline variant of PDAF as discussed in the introduction and the next paragraph. We clarified this in the text.

Changes in Text:
Lines: 401-405

"The presented implementation can be summarized by three main aspects that will be discussed in this section: the online variant of PDAF, re-use of CLM5 ensemble mode, and the TSMP framework. The online variant of PDAF performs data assimilation in the main memory during runtime by coupling the model and PDAF in a single executable."

**Figures: Add subplot numbers (e.g., a), b))**

We added subplot numbers.

See figures 3 to 7.

**Figure 2: "In the diagram NMLST means namelist, SIM means simulation process, HIST means history file output, PID means PDAF identification number." – this should as legend in Figure.**

We added a legend with the shorthand.

[Figure]

Figure 2: Schematic overview of CLM5 ensemble mode (left side) and CLM5+PDAF (right side) communication initialization and process flow. The schema highlights the addition of communication for all ensemble members through the PDAF communication model.

**Figure 3 caption: red (solid line), light green (dotted line).**

We corrected the caption.

Changes in Text:
Line: 628-634

"Figure 3: Time series of the monthly averaged soil water content (SWC) from 2009 to 2018 at the three different depths and for each simulation scenario. The subplots A), B), and C) represent the three depths 5cm, 20cm, and 50cm respectively. The red (solid line) shows observational data. The light green (dotted line) shows open loop simulation results. The blue (dash-dotted line) shows results for data assimilation of state variables. The purple (dashed line) shows results for the assimilation of states and updating of parameters. The dark green (dashed line) shows results for assimilation of states and updating of parameters including organic matter."